# Modulation of metal species as control point for Ni-catalyzed stereodivergent semihydrogenation of alkynes with water

Yuanqi Wu[1], Yuhui Ao[1], Zhiming Li [2] ✉, Chunhui Liu[3], Jinbo Zhao[1], Wenyu Gao[1], Xuemeng Li[1], Hui Wang[1], Yongsheng Liu[1] & Yu Liu [1] ✉

A base-assisted metal species modulation mechanism enables Ni-catalyzed stereodivergent transfer semihydrogenation of alkynes with water, delivering both olefinic isomers smoothly using cheap and nontoxic catalysts and additives. Different from most precedents, in which $E$-alkenes derive from the isomerization of $Z$-alkene products, the isomers were formed in orthogonal catalytic pathways. Mechanistic studies suggest base as a key early element in modulation of the reaction pathways: by adding different bases, nickel species with disparate valence states could be accessed to initiate two catalytic cycles toward different stereoisomers. The practicability of the method is showcased with nearly 70 examples, including internal and terminal triple bonds, enynes and diynes, affording semi-hydrogenated products in high yields and selectivity.

Divergent catalysis as a particularly appealing strategy from both academic and practical perspectives allows convenient control over selectivity towards different terminal products starting from the same material[1–7]. Predictably, it would be more beneficial for the distinction of reactivity and selectivity if the two target molecules are achieved in separate mechanistic pathways, which generally requires employment of different catalysts to initiate diverse catalytic cycles. Therefore, it would be mechanistically interesting and also operationally practical to develop novel strategies in which different catalytic species could be generated from the same catalyst precursor by simple adjustment of the reaction factors, leading to different products with high selectivity in two independent catalytic cycles.

Transition metal-catalyzed stereodivergent hydrogen transfer of alkynes to produce both $Z$- and $E$-olefins have attracted remarkable interest in recent years[8–20]. Most pioneering examples actualize this transformation by regulation of catalytic systems to realize a $Z$ to $E$ isomerization process at the late stage (Fig. 1, above). For instance, Moran et al. showed that Ni-catalyzed transfer hydrogenation (TH) of alkynes with $HCO_2H$ selectively afforded $Z$-olefins, which isomerized to $E$-isomers by adding triphos ligand[8]. Another catalyst-modulated system was disclosed by Liu and coworkers in 2016, in which both isomers could be achieved using Co catalysts supported with specified

bidentate ligands. The isomerization of $Z$-alkenes was suppressed by introducing bulky ligand due to the sterically unfavored coordination and insertion processes[9]. Recently Mei et al. reported that Pd-catalyzed semihydrogenation of alkynes with $H_2O$ delivered $cis$-olefins at room temperature in $CH_3CN$, while isomerization of the double bond towards $trans$-olefins was facilitated at 80 °C in DMF[10]. Mechanistically, $E$-alkenes in the majority of reported strategies originate from the $Z$-isomer, requiring subtleness of the reaction conditions and the steric or electronic properties of the substrates. Therefore, mechanistically orthogonal stereodivergent semi-reduction of alkynes to both olefinic isomers, in which $E$-alkenes are generated directly from alkynes instead of the isomerization from $Z$-alkenes, is undoubtedly meaningful in both academic and practical perspectives. We envisioned that modulating the catalytic species at an early stage might initiate independent profiles to deliver both isomers in orthogonal manners (Fig. 1, bottom). Ideally, several issues should be addressed: (a) independent catalytic cycles should be initiated by simple adjustment of the reaction factor(s) to enable high yield and stereoselectivity[8–14,21,22]; (b) nonprecious metals and ligands without toxic additives would be more favorable[23–27]; (c) water is the first choice of the hydrogen donor for TH process[10,28–31]; (d) alkynes with various substituents should be hydrogenated in high yield and stereoselectivity in mild conditions, and

[1]Jilin Provincial Key Laboratory of Carbon Fiber Development and Application, College of Chemistry and Life Science, Advanced Institute of Materials Science, Changchun University of Technology, 130012 Changchun, PR China. [2]Department of Chemistry, Fudan University, 200438 Shanghai, PR China. [3]College of Chemical and Materials Engineering, Xuchang University, 461000 Xuchang, PR China. ✉e-mail: zmli@fudan.edu.cn; yuliu@ccut.edu.cn

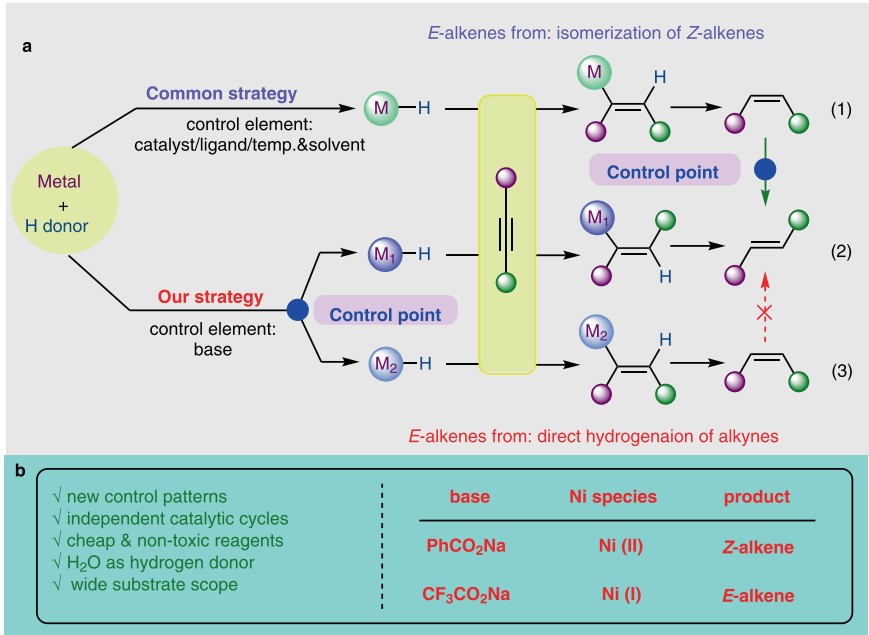

**Fig. 1 | Transition metal-catalyzed stereodivergent TH of alkynes to alkenes. a** Regulation of stereoselectivity in transition metal-catalyzed stereodivergent TH of alkynes. **b** Modulation of stereoselectivity in this work.

over-reduction to saturated alkanes need to be avoided[32,33]. Pioneered by previous Ni-catalyzed alkyne hydrogenation[8,34–36], we launched a project with nickel catalysts to address the above challenges. After laborious trials, we realize a Ni-catalyzed stereodivergent TH of alkynes with water in an innovative controlling mode, in which the key to the success of modulation is the judicious inclusion of the base. Notably, unlike most existing reports, formation of *trans*-olefins is unrelated to the isomerization of *cis*-olefin. Mechanistic investigations suggest that base modulated the valence state of active nickel species derived from the same simple pre-catalyst. Consequently, the isomers are achieved independently in completely disparate catalytic pathways: the in situ formed Ni(II) species delivered *Z*-alkenes, while the Ni(I) species selectively afforded *E*-alkenes as final products.

## Results

### Optimization of the reaction conditions

We initiated our exploration by evaluating the transfer hydrogenation of the model substrate **1a** with simple nickel sources and 2,2′-bipyridine ligands (Table 1). The first obstacle to overcome is the activation of the inert $H_2O$ molecule in our nickel catalyst system[37–39]. Gratifyingly, boron reagents showed unique effect, and the alkenes were obtained in high yield and selectivity using $Na_2CO_3$ as base. $B_2pin_2$ turned out to be more efficient than other diboron compounds such as $B_2(OH)_4$, $B_2cat_2$ and $B_2neop_2$ (Supplementary Table 1)[40–42]. Although diboron compounds were found to be capable of activating water in Pd-catalyzed systems[40–44], including hydrogenation of unsaturated C-C bonds to saturated alkanes[40], it is, to our knowledge, the first case for such activation effect in Ni catalyst systems. Notably, *E*-alkene **3a** was formed as the major isomer, and over-reduced alkane product was not observed. Solvents turned out to exert a profound influence on the reactivity (Supplementary Table 1), and 72% yield of alkenes were obtained with 11/89 isomeric ratio in DMF (entry 1). Decorating the bipyridine ligand with electron-withdrawing ester groups totally suppressed the reactivity (entry 2). Subsequent screening of other bipyridine derivatives as well as phenanthroline ligands **L3-L6** provided comparatively inferior results to 2,2′-bipyridine (entries 3-6). Systematic screening of nickel catalyst, ligand, base, boron and water (Supplementary Tables 2 and 3) showed that base exerted an

unexpected, yet decisive role in the control of selectivity. As shown in Table 1, the reaction was evidently inclined to *E*-selectivity by $K_2CO_3$, NaOH and $CF_3CO_2Na$, with the later showing the best result, affording **3a** in 84% isolated yield and 6/94 *Z/E* ratio (entries 7–9). Interestingly, a slant to *Z*-selectivity was shown with $CH_3CO_2Na$, providing **2a** with 69/31 *Z/E* ratio (entry 10). Organic bases such as DABCO and $Et_3N$ were also tested, and *E*-alkene **3a** was delivered as the major product (entries 11 and 12). The catalyst loading could be lowered to 5 mol% with no erosion of the yield or selectivity (entry 13). The reactivity was almost totally shut down at a lower temperature of 60 °C (entry 14). The alkyne **1a** was untouched at 40 °C, leaving all starting material recovered (entry 15). In contrast, comparable results were observed at higher temperatures (entries 16 and 17).

The above results inspired us to further proceed with other bases aiming at the optimization for *Z*-selective transfer semihydrogenation of **1a**. As shown in Table 2, $CH_3CO_2K$ and $CH_3CO_2Cs$ acted similarly as $CH_3CO_2Na$, indicating that metal ions are not responsible for the selectivity reversal (entries 1 and 2). Only moderate selectivity was achieved when $HCO_2Na$ was added (entry 3). To our delight, $PhCO_2Na$ gave a promising result, providing the final olefins in 80/20 selectivity (entry 4). Again, dicarboxylate ligand **L2** showed dramatically decreased reactivity (entry 5). In contrast, 4,4′-dimethoxy-2,2′-bipyridine **L3** improved the selectivity to 93/7 (entry 6). Ligands **L4** and **L5** bearing methyl groups at 3,3′- or 4,4′-positions both gave slightly reduced selectivity than **L3** (entries 7 and 8). When the loading of the catalyst and base were reduced, alkenes were retrieved in slightly improved yield and selectivity (entries 9 and 10). Contrary to *E*-selective system (Table 1, entry 14), the reaction could still proceed smoothly at a lower temperature, albeit **1a** was partially recovered (entry 11). Performing the reaction at higher temperatures resulted in poorer selectivities (entries 12 and 13).

### Mechanistic investigations

Several questions deserve exploration to better understand this unprecedented system: (a) is water in the system indeed the hydrogen donor? (b) are alkenes generated from hydrometallation of in situ formed Ni-H species or hydrolysis of vinyl boron compounds? (c) does isomerization of *Z*-olefins take effect similarly as most precedents to

**Table 1 | Optimization for the *E*-selective transfer semihydrogenation of 1a[a]**

| Entry | Ligand | Base | T/°C | Yield/%[e] | 2a/3a[e] |
|-------|--------|------|------|-----------|----------|
| 1 | L1 | $Na_2CO_3$ | 80 | 72[d] | 11/89 |
| 2 | L2 | $Na_2CO_3$ | 80 | 0 | - |
| 3 | L3 | $Na_2CO_3$ | 80 | 62 | 19/81 |
| 4 | L4 | $Na_2CO_3$ | 80 | 70 | 18/82 |
| 5 | L5 | $Na_2CO_3$ | 80 | 64 | 25/75 |
| 6 | L6 | $Na_2CO_3$ | 80 | 71 | 12/88 |
| 7 | L1 | $K_2CO_3$ | 80 | 42 | 22/78 |
| 8 | L1 | NaOH | 80 | 30 | 33/67 |
| 9 | L1 | $CF_3CO_2Na$ | 80 | 84[d] | 6/94 |
| 10 | L1 | $CH_3CO_2Na$ | 80 | 90 | 69/31 |
| 11 | L1 | DABCO | 80 | 74 | 12/88 |
| 12 | L1 | $Et_3N$ | 80 | 62 | 11/89 |
| 13[b] | L1 | $CF_3CO_2Na$ | 80 | 83[d] | 5/95[f] |
| 14 | L1 | $CF_3CO_2Na$ | 60 | 4[e] | - |
| 15 | L1 | $CF_3CO_2Na$ | 40 | - | - |
| 16 | L1 | $CF_3CO_2Na$ | 100 | 78 | 6/94 |
| 17 | L1 | $CF_3CO_2Na$ | 120 | 80 | 7/93 |

[a]Reactions were performed with **1a** (0.15 mmol), $NiBr_2$ (10 mol%), **L1** (22 mol%), base (2.0 equiv.), $B_2Pin_2$ (3.0 equiv.), $H_2O$ (3.0 equiv.), DMF (2 mL), 80 °C, 10 h.
[b]5 mol% of $NiBr_2$, 11 mol% of **L1**.
[c]Determined by crude $^1H$ NMR.
[d]Isolated yield.
[e]Only Z-alkene product **2a**.
[f]Determined by GC.

afford *E*-olefins? (d) what are the roles of the bases in modulation of the reaction outcomes? To answer these questions, a series of mechanistic studies were carried out. Firstly, deuterium-labeled experiments were conducted (Fig. 2a). The deuterium was incorporated into both the 1,2-olefinic positions of **2a'** and **3a'** with $D_2O$ instead of $H_2O$ under both standard conditions (equations (1) and (2)). Similar results were also observed for unsymmetric alkynes **1bb** and **1i**, with the former leading to even higher deuterations (equations (7) and (8)). In contrast, there was no sign of deuteration on the products using DMF-$d_7$ as solvent (equations (3) and (4)). When the reactions of **1a** using $D_2O$ were placed in hydrogen atmosphere, comparative deuterium isotopic contents as in argon were observed (equations (5) and (6)), proving that releasing of $H_2$ and consequent hydrogenation was not involved in the catalytic pathway. Control experimental studies of vinylboron reagents **4**, **5** and diborylated vinyl derivatives **4'**, **5'** were respectively performed under both reaction conditions with 0, 1.0, 2.0, and 3.0 equiv. of $B_2pin_2$[45–48]. Olefin products **2g** and **3g** were not detected (Fig. 2b, (equations (9), (10), (11) and (12)). This, together with the reactions under $H_2$ atmosphere, indicated that Ni-H species were formed between the nickel pre-catalyst and $H_2O$ assisted by $B_2pin_2$, which would deliver alkenyl nickel intermediates to accomplish the catalytic cycle.

To deeper understand the process of selective semi-reduction, the kinetic behavior of the reaction system was monitored (Fig. 2c). The kinetic profile of *Z*-selective transfer semihydrogenation system showed that the concentration of **2a** increased gradually throughout the reaction period, staying closely aligned with the conversion of **1a**.

After 5 h, **3a** began to show up until the yield reached 6% (Fig. 2c, left). We postulate that the small amount of *E*-alkene in this system was generated from isomerization of the *Z*-product, which was suppressed in the initial period due to competitive coordination of alkyne **1a** with the metal center. Consumption of most **1a** after 5 h left space for the coordination of **2a** for the subsequent isomerization process, which still need **1a** as auxiliary since the selectivity remained unchanged after disappearance of **1a**. The *E*-selective reaction profile with $CF_3CO_2Na$ as base clearly indicated the nonexistence of *Z/E* isomerization (Fig. 2c, right). Approximately 6% of *Z*-alkene was already formed at the early stage of the reaction, which maintained in this level until **1a** was completely converted. The concentration of **3a** increased gradually, which was independent with the amount of **2a**. The kinetic isotopic effect ($k_H/k_D = 1.68$) was observed when $H_2O$ was replaced by $D_2O$ in the *Z*-selective reactant stream (Supplementary Figure 1), and a kinetic isotopic effect of 1.08 was also obtained in the *E*-selective reduce system (Supplementary Figure 2), indicating that activation of $H_2O$ molecule delivering Ni-H species might not be involved in the rate-determining step in both selective hydrogenations.

To further verify the above inferences, a series of control experiments were conducted. When *Z*-alkene **2a** was put in both standard conditions, only less than 5% of *E*-alkene was detected (Supplementary Fig. 3, equations (1) and (2)), demonstrating the reluctance of the *Z/E* isomerization in these conditions. Elevating the reaction temperature showed a beneficial effect for the isomerization, which was promoted to 13% by heating **2a** at 120 °C under the *Z*-selective condition (Supplementary Fig. 3, equation (3)). Consistently, the reaction of **1a** at

**Table 2 | Optimization for the *Z*-selective transfer semihydrogenation of 1a[a]**

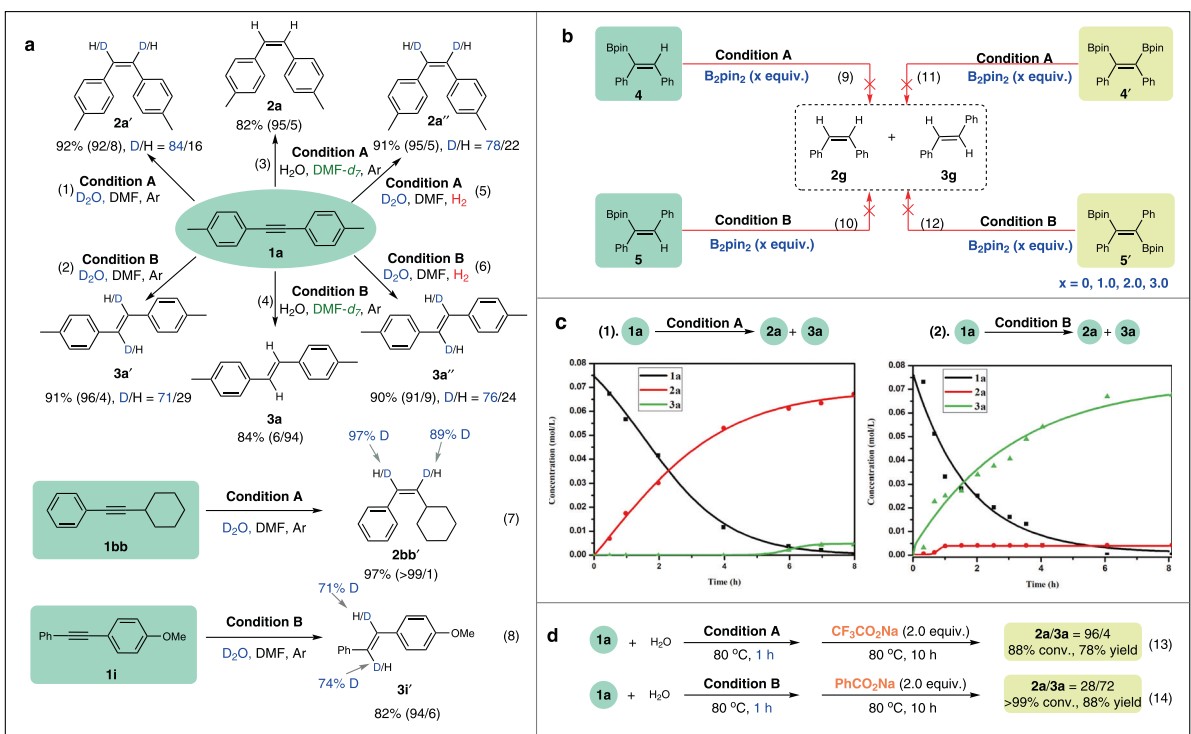

| Entry | Ligand | Base | T/°C | Yield/%[c] | 2a/3a[c] |
|---|---|---|---|---|---|
| 1 | **L1** | CH₃CO₂K | 80 | 87 | 70/30 |
| 2 | **L1** | CH₃CO₂Cs | 80 | 97 | 72/28 |
| 3 | **L1** | HCO₂Na | 80 | 85 | 56/44 |
| 4 | **L1** | PhCO₂Na | 80 | 93 | 80/20 |
| 5 | **L2** | PhCO₂Na | 80 | 22 | - |
| 6 | **L3** | PhCO₂Na | 80 | 83 | 93/7 |
| 7 | **L4** | PhCO₂Na | 80 | 90 | 90/10 |
| 8 | **L5** | PhCO₂Na | 80 | 83 | 91/9 |
| 9[b] | **L3** | PhCO₂Na | 80 | 90[d] | 94/6 |
| 10[b,e] | **L3** | PhCO₂Na | 80 | 90[d] | 94/6 |
| 11[f] | **L3** | PhCO₂Na | 60 | 59 | 94/6 |
| 12 | **L3** | PhCO₂Na | 100 | 91 | 90/10 |
| 13 | **L3** | PhCO₂Na | 120 | 87 | 86/14 |

[a]Reactions were performed with **1a** (0.15 mmol), NiBr₂ (10 mol%), **L** (22 mol%), base (2.0 equiv.), B₂Pin₂ (3.0 equiv.), H₂O (3.0 equiv.), DMF (2 mL), 80 °C, 12 h.
[b]5 mol% of NiBr₂, 11 mol% of **L3**.
[c]Determined by crude ¹H NMR.
[d]Isolated yield.
[e]1.0 equiv. of PhCO₂Na.
[f]The conversion of **1a** was 68% after 15 h.

**Fig. 2 | Mechanistic experiments. a** Deuterium labeling experiment. **b** Control experimental studies of vinylboron reagents. **c** Kinetic profiles of the reaction systems. **d** Competitive control experiments of the bases.

120 °C under this condition afforded the corresponding olefinic products in 86/14 selectivity (Supplementary Fig. 3, equation (4)), compared with 93/7 at 80 °C.

The color changes between the two reaction systems were significantly different. As shown in Supplementary Fig. 4, the Z-selective system seemed turbid and beige at the very beginning, which turned to light brown after several minutes and got darker later. The color changed to tan-yellow gradually in about one hour and became lighter to milk-white after another one hour, which remained till the end. A completely different visual appearance mutation was observed for the E-selective system, which looked transparent black and got darker quickly at the very early stage. Interestingly, as soon as the reaction was over as monitored of the crude mixture, the color changed to bright yellow immediately, which could be regarded as a simple hint for the complete of the reaction. We postulate that the dark color ascribes to the coordination of the triple bond to the metal center, which was terminated promptly once alkynes were exhausted[33]. The distinction in colors of the two systems indicates that different nickel species might be involved, leading to the corresponding olefinic products in totally unrelated pathways. The color variation of the control experiments on base was quite similar to the above observation (Supplementary Fig. 4, bottom): the initial pale green color changed to tint of turbidity yellow and clarify black color separately after addition of PhCO$_2$Na and CF$_3$CO$_2$Na, respectively, indicating the formation of different nickel species was modulated with the choice of base.

Competitive control experiments of the bases were conducted to further illustrate their functions (Fig. 2d). After the standard Z-selective mixture using PhCO$_2$Na was stirred for 1 h, another 2.0 equivalent of CF$_3$CO$_2$Na was added, and no apparent influence on the reaction outcomes was observed (equation (13)). By contrary, a worse selectivity was caused by addition of PhCO$_2$Na into the E-selective system (28/72 vs 4/96) (equation (14)).

All the mechanistic insights and the visual phenomenon pointed to distinct catalytic pathways for the two reaction systems, inspiring us to further inquire whether different metal species were taking effect inherently. To detect whether nanoparticles were involved in our Ni-B-H$_2$O system, general mercury drop experiments were performed[41,43]. The yield or selectivity was not affected in either system (Supplementary Fig. 5, equations (1) and (2)), excluding heterogeneous catalytic pathway. Despite the failure in capture of metallic intermediates, electron paramagnetic resonance (EPR) analyses provided clues on the active nickel species and the base effect. As shown in Fig. 3a (2), strong EPR signals were observed in the E-selective mixture, indicating the formation of Ni(I) or Ni(III) species[49–52]. The signals of such Ni species could not be found at ambient temperature, which is in accordance with our experimental observations that semihydrogenations of 1a were not permitted at rt (Supplementary Table 3, entry 25). In contrast, EPR active species was not observed in Z-selective system (Fig. 3a (1)) that features a Ni(0)/Ni(II) catalytic cycle. In agreement with the competitive experiments of bases (equation (14)), the EPR signals for the reactions using CF$_3$CO$_2$Na as base were markedly weakened after the addition of PhCO$_2$Na (Fig. 3a (3)). In line with the fact that use of HCO$_2$Na as base gave an almost 1:1 ratio of the Z- and E-alkenes (Table 2, entry 3), the EPR signal of the system with HCO$_2$Na was less significant than that with CF$_3$CO$_2$Na (Fig. 3a (4)), but much more significant than that of PhCO$_2$Na system (Fig. 3a (1)). EPR test results in the absence of alkyne and water are shown in Fig. 3a (5), which is in parallel to the studies in presence of starting materials: no signal was observed when PhCO$_2$Na was employed, while a strong EPR signal showed up with CF$_3$CO$_2$Na as base, demonstrating that different Ni species are accessed at early stage without the participation of alkynes. In addition, we also analyze the mixture of NiBr$_2$ and Ni(cod)$_2$, which would generate Ni(I) species in situ (Fig. 3a (6))[53,54]. The similar signal compared with the system using CF$_3$CO$_2$Na further demonstrate the involvement of Ni(I) species in the E-selective protocol.

Further control experiments were carried out to verify the key role of Ni(I) species in the E-selective hydrogenation process. As shown in Fig. 3b, in situ formed Ni(I) species by mixing NiBr$_2$ and Ni(cod)$_2$ resulted in olefin 3a with E-configuration as major product (40% yield, 18/82 Z/E). Noteworthy, the reaction was totally suppressed in this condition without base (Supplementary Table 3, entry 24), which is another evidence for the participation of CF$_3$CO$_2$Na to deliver Ni(I) species. On the other hand, the Z/E ratio dropped appreciably when Mn or Zn was added in the Z-selective condition (Fig. 3c), which might due to the generation and competitive act of Ni(I) species. Furthermore, the reactivity in condition A was suppressed when Zn or Mn was used instead of B$_2$Pin$_2$ (Fig. 3d), indicating that B$_2$Pin$_2$ not only interacts with bases to deliver active Ni species in this hydrogenation system, but also exists as an activator of water.

Although more experimental supports are awaited to uncover the detailed mechanism (initial NMR studies on the mechanism, see Supplementary Fig. 6), a general scenario could be delineated based on the above results and related literatures[34,35,55–63] (Fig. 3e): NiBr$_2$ would interact with the bases firstly, delivering carboxylates carrying different counter anions. The difference in electronic properties between the benzoate and the trifluoroacetate endows them with distinct reactivities towards B$_2$pin$_2$. Organic bases such as DABCO and Et$_3$N inclined to deliver E-olefins (Table 1, entries 11 and 12), which is in consist of our proposal that counter anion from the base was not equipped to the metal center in the Z-selective catalytic cycle[64]. Consequently, Ni(II) species C is generated directly from the benzoate B and B$_2$pin$_2$ for Z-selective catalytic cycle. Activation of H$_2$O molecule delivers Ni(II)-H species D, which undergo syn-addition to the triple bond to afford alkenyl Ni(II) intermediate E. Participation of another H$_2$O molecule releases the cis-olefin and regenerate C with the assistance of B$_2$pin$_2$. Based on the kinetic experiments, coordination and insertion of the Z-alkene to the Ni-H species assisted by alkyne precursor would occur in the late stage of the reaction, followed by isomerization process resulting in slight stereo-impurity. We propose that isomerization of a vinyl Ni(I) species is responsible for the E-selectivity observed in this approach, the specific oxidation state at Ni could provide an opportunity for isomerization[55,56]. At the beginning of the cycle, Ni(II) species H might be generated firstly from nickel trifluoroacetate G and B$_2$pin$_2$, which furnishes Ni(0) species I in a reductive elimination step. Comproportionation between H and I occurs instantly, forging Ni(I) species J to initiate the catalytic cycle[53,54]. Activation of H$_2$O molecule would deliver Ni(I)-H species L, followed by insertion of alkyne leading to vinyl Ni(I) intermediate M, which may undergo isomerization[56] to E-alkenyl nickel intermediate N. Thermodynamically more stable product 3 is generated by hydrolysis of N, and the acquired nickel hydroxide O could be transformed back to Ni(I) species J in the aid of B$_2$pin$_2$.

Density functional theory (DFT) calculations were carried out to investigate the remarkably different impact of PhCO$_2$Na and CF$_3$CO$_2$Na on Ni species[65–68]. The reaction free energy profiles are shown in Fig. 4. The Ni(II) precursor A reacted with PhCO$_2$Na to afford nickel benzoate B, which is exergonic by 44.7 kcal/mol (Fig. 4a). The activation free energy barrier for one-ligand exchange of B towards PhCO$_2$-Ni(II)-BPin C is 25.2 kcal/mol. The activation of C with H$_2$O molecule requires a 14.3 kcal/mol of activation free energy en route to Ni(II)-H species D. Although the transition state of further ligand exchange from C to P could not be located, the process from P to Q was unfavored due to 29.9 kcal/mol of activation energy barrier, supporting our proposal that the catalytic cycle proceeds through benzoate D. As for the catalytic system using CF$_3$CO$_2$Na as base, LNiBr$_2$ A firstly reacted with CF$_3$CO$_2$Na to give Ni(II) species G, which is exergonic by 48.4 kcal/mol. One-ligand exchange with B$_2$Pin$_2$ producing CF$_3$CO$_2$-[NiL]-BPin H is endothermic with 13.4 kcal/mol, and the energy barrier is 29.4 kcal/mol, which is in parallel with the experimental result that the reactivity was almost totally suppressed at a lower temperature of 60 °C (Table 1,

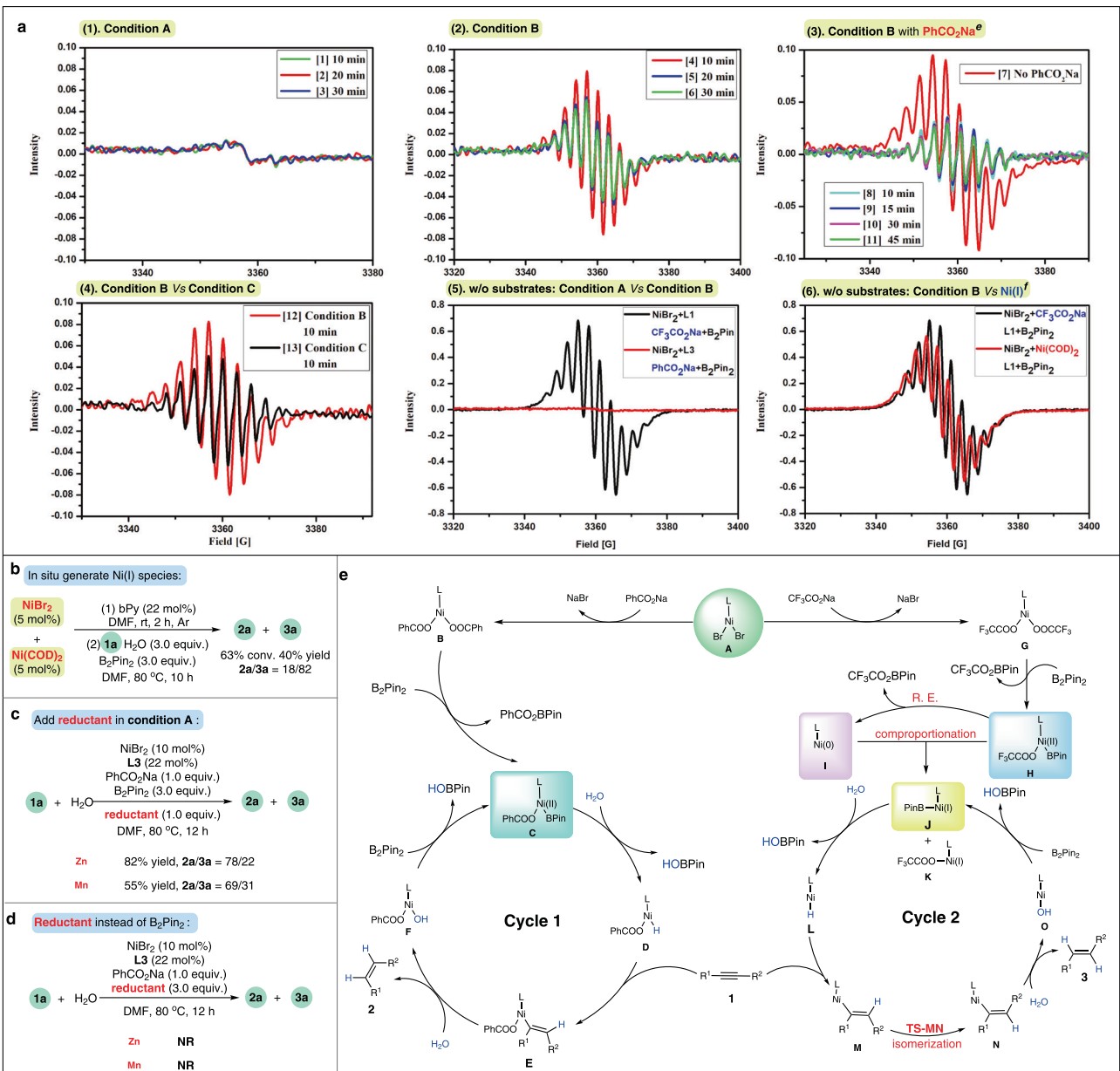

**Fig. 3 | Further investigations on catalytic pathways. a** EPR analyses. **b** Control experiment of Ni(I) species. **c** and **d** Control experimental studies of reductant in the *Z*-selective condition. **e** Proposed catalytic cycles.

Entry 14). Although we were unable to locate the transition state for further ligand exchange from **H** to **P**, the notably higher reaction Δ*G* of two-ligand exchange makes it more unlikely. For the one-ligand exchange pathway, subsequent reductive elimination and compro-portionation process are exergonic, leading to Ni(I) species **J** and **K** to initiate the *E*-selective catalytic cycle. The activation free energy barrier for the reaction of **H** with $H_2O$ molecule en route to Ni(II)-H species **R** is nearly 10 kcal/mol higher than that of the reductive elimination. Besides, **J** + **K** is lower than **R** in the potential energy surfaces, basically excluding the participation of **R** in the catalytic cycle.

## Substrate scope

The synthetic practicability of this system was sufficiently embodied in the functional group compatibility investigations. In Fig. 5a, the *Z*-selective semi-reduction of various alkynes **1** using $PhCO_2Na$ as base is summarized. This reaction proceeded successfully toward substituted diarylethynes bearing a diverse set of substituents. Specifically, sub-strates bearing methyl or *tert*-butyl groups at *p*- or *m*-positions all worked smoothly under the standard conditions (**2a**–**2d**), as well as hindered isopropyl (**2e**) or phenyl (**2f**) groups located in the *ortho*-position of the aryl terminus, suggesting the insensitivity of the system to steric effect. Electron-donating methoxy substituent was well accommodated, and the diaryl alkenes were generated in high yields and selectivity (**2h**, **2i** and **2j**). Amino functional group **2k** was no exception, well tolerated in this catalytic transfer semihydrogenation process. *Z*-olefins with electron-withdrawing trifluoromethyl (**1l**), cyano (**1m**, **1n**), ester (**1o**) and acyl (**1p**) groups could also be achieved uneventfully. Fluoro- and chloro-containing products (**2q**-**2t**) were furnished from the corresponding alkynes, leaving space for further functionalization. Arylalkyne **1u** bearing hydroxyl group provided the

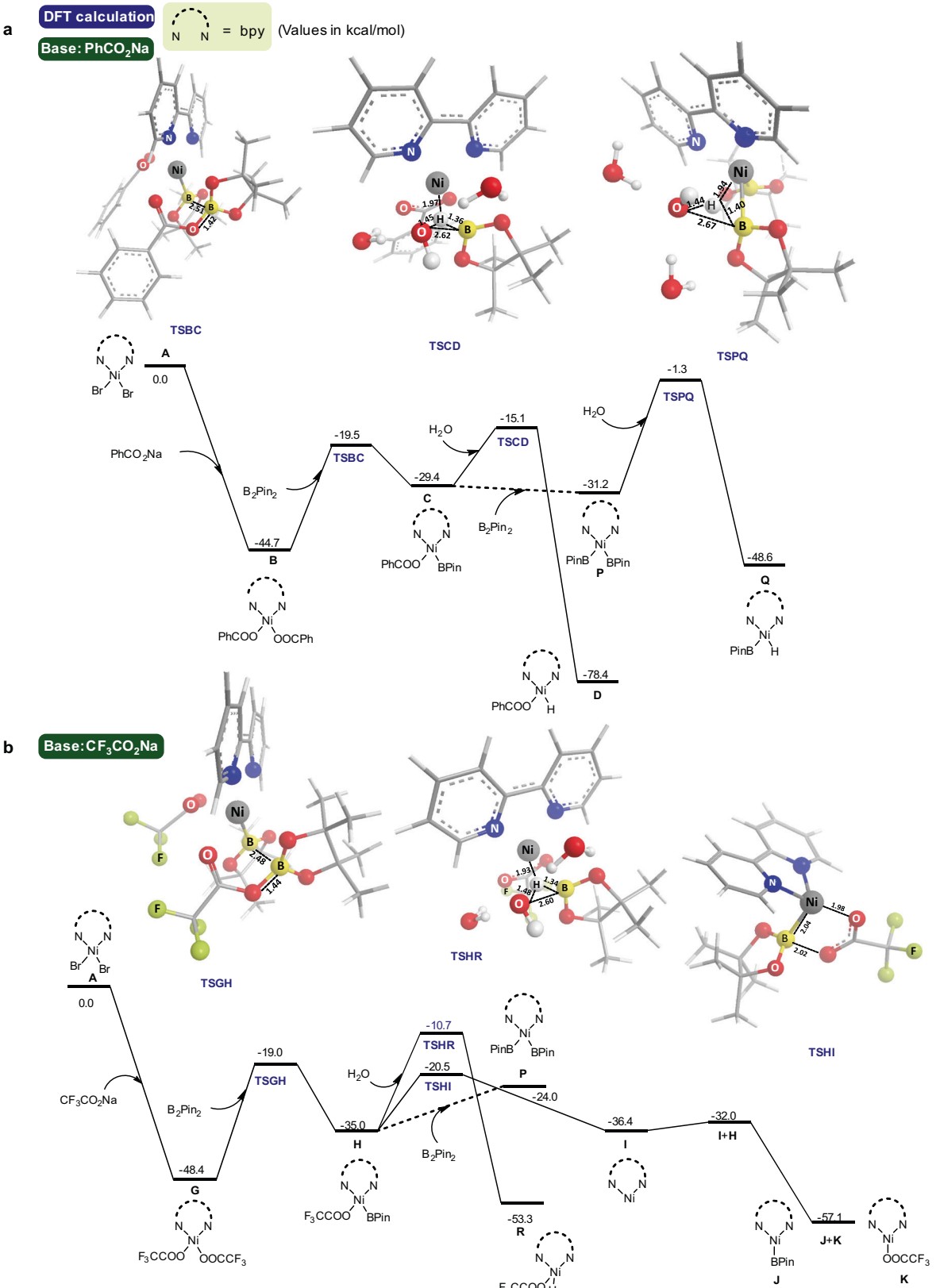

**Fig. 4 | DFT calculations for reaction mechanism. a** The Free-energy reaction profile of *Z*-selective catalytic cycle. **b** The Free-energy reaction profile of *E*-selective reaction system.

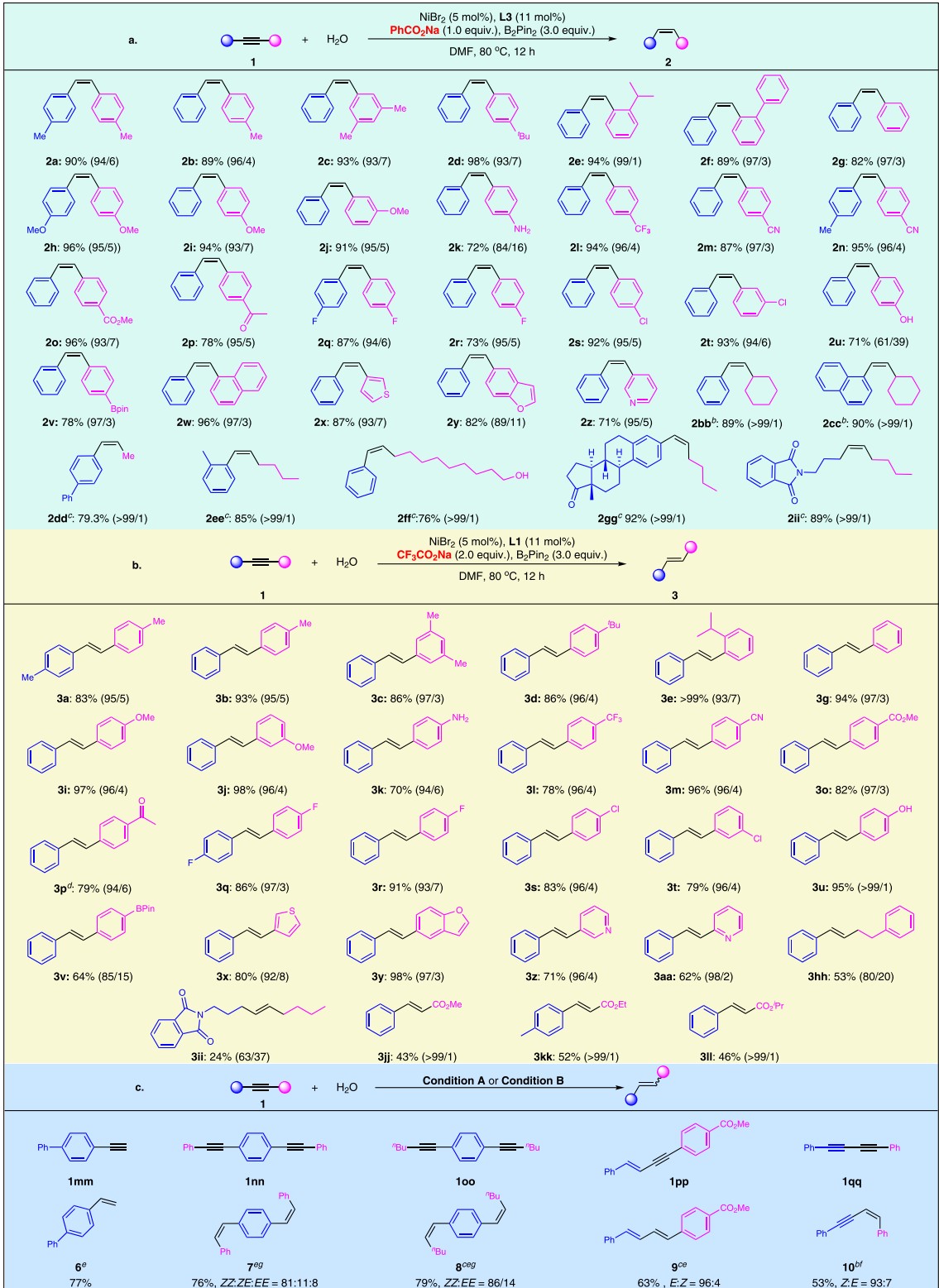

**Fig. 5 | The scope of the transfer semihydrogenation system. a** Scope of TH of alkynes to *Z*-alkenes with water. **b** Exploration of substrate scope in *E*-selective condition. **c** Substrate scope with respect to terminal triple bonds, enynes, and diynes.

alkene product with 61/39 *Z/E* ratio. The relatively lower selectivity might be caused by isomerization of *Z*-olefin **2u**, since the *Z/E* ratio of the olefins decreased from 86/14 to 53/47 by heating in DMF at 80 °C. The alkyne derivative containing Bpin-substituent was well tolerated, providing alkene **2v** in 78% yield and 97/3 *Z/E*. The generality of the system was further showcased by the tolerance of naphthyl (**2w**) and heterocycles including thienyl (**2x**), benzofuryl (**2y**) and pyridyl motifs (**2z**), particularly the latter, considering pyridinyl ligands were used in our catalytic system. Moreover, running in a longer reaction time or higher temperature, alkynes carrying both naphthenic and linear alkyl terminuses could be reduced to the corresponding olefinic products efficiently (**2bb**–**2hh**). Notably, only *Z*-alkenes were formed specifically from the alkyl substrates, supporting our previous deduction that the *E*-alkenes in the *Z*-selective conditions might derive from the isomerization process, which was sluggish for alkyl alkenes due to their weak coordinating ability to the metallic species. The compatibility of the system was further underlined by successful involvement of unprotected primary OH group (**2ff**), which was unaffected under the catalytic conditions. Natural product derived alkyne with estrone skeleton proceeded smoothly, and the desired product **2gg** was furnished in excellent yield and selectivity. Finally, internal alkyne **1ii** bearing 1,2-dialkyl substituents also gave high yield and perfect stereoselectivity.

A survey on the substrate scope was performed next to demonstrate the robustness of the *E*-selective TH process using $CF_3CO_2Na$ as base (Fig. 5b). Similar as the former system, diaromatic internal alkynes with a wide range of functional groups such as methyl (**1a-1c**), *tert*-butyl (**1d**), isopropyl (**1e**), methoxyl (**1i**, **1j**), amino (**1k**), trifluoromethyl (**1l**), cyano (**1m**), ester (**1o**) and acyl (**1p**) and halogen substituents (**1q-1t**) were all hydrogenated to the desired *trans*-alkenes uneventfully. Alkyne bearing a hydroxyl on the aromatic ring worked well to furnish the desired product **3u**. Bpin group might interact competitively with the active species in this Ni-B hydrogenation system, thus olefin **3v** was achieved in inferior selectivity (85/15 *E/Z*). Heteroaromatic rings including thienyl (**1x**), benzofuryl (**1y**) and pyridyl (**1z**, **1aa**) substituents were compatible again, delivering the alkenyl heterocycles selectively. *Trans*-selective transfer semihydrogenation of alkyl acetylenes turned out to be challenging: monoaryl alkyne **1hh** was reduced to **3hh** in moderate yield and selectivity, while dialkyl alkyne **1ii** delivered **3ii** in much more inferior result. Propargylic esters were transformed to *E*-olefins (**3jj**–**3ll**) as single isomers in moderate yields and excellent selectivity. Consistent with the previous observation, for all the *E*-selective experiments, a mutation of color from black to bright yellow was observed as soon as the reaction finished.

Finally, we tested the reactivity of terminal alkynes, which are more inclined to over-reduction. As shown in Fig. 5c, alkene **6** was obtained in high yield in *Z*-selective conditions from **1mm**, and saturated ethyl product was not observed. The condition could also be extended to diynes **1nn** and **1oo**, with both triple bonds being hydrogenated in high selectivity. Interestingly, the reaction of conjugated enyne **1pp** in *Z*-selective conditions afforded diene **9** with *E*-configuration as the major product. On the contrary, *Z*-enyne **10** was obtained in high selectivity when 1,3-diyne **1qq** was loaded in *E*-selective conditions.

## Discussion

In conclusion, we have disclosed an unprecedented Ni-catalyzed stereodivergent transfer semihydrogenation of acetylenes with water. The configuration of the olefinic products was controlled by the choice of bases, which were demonstrated to influence the valence states of the catalytic nickel species. Consequently, *E*-alkenes were achieved independently from the direct reduction of alkyne precursors instead of isomerization of the *Z*-isomers. The strategy also features use of cheap catalysts and nontoxic reagents, and compatibility with an assortment of alkynyl substrates such as internal and terminal alkynes, 1,3-enynes and diynes. Besides its significance in semihydrogenation of

alkynes, we believe that the mechanistic insights would lead to better understanding of the performance of nickel species, and also pave the way to further exploration of the other transition metal catalyst systems. Further pursuit including the development of the catalytic strategy and also detailed mechanistic studies are ongoing in our laboratory.

## Methods

### General procedure for *Z*-selective transfer semihydrogenation of alkynes 1

To a dry sealed tube were added alkyne **1** (0.3 mmol), $NiBr_2$ (3.3 mg, 0.015 mmol, 5 mol%), **L3** (7.1 mg, 0.033 mmol, 11 mol%), $PhCO_2Na$ (43.2 mg, 0.3 mmol, 1.0 equiv.) and $B_2pin_2$ (228.5 mg, 0.9 mmol, 3.0 equiv.). The flask was evacuated and refilled with argon, followed by the addition of $H_2O$ (16.2 μL, 0.9 mmol, 3.0 equiv.) and DMF (4 mL). The mixture was stirred at 80–100 °C for 8–30 h until the reaction was completed as monitored by TLC. The resultant solution was diluted with ethyl acetate, washed with HCl aqueous solution (1 M) and concentrated in vacuum. The mixture was detected by GC directly or after simple filtration in some cases to determine the *Z/E* ratio. The crude product was purified by chromatography on silica gel (300–400 mesh), eluted with petroleum ether with 0–20% of ethyl acetate to give alkene product. Careful column chromatography was able to partially deliver the major product in a pure form to provide precise NMR spectra of the major product. The overall isolated yield was calculated based on the combination of all parts.

### General procedure for *E*-selective transfer semihydrogenation of alkynes 1

To a sealed tube were added alkyne **1** (0.3 mmol), $NiBr_2$ (3.3 mg, 0.015 mmol, 5 mol%), **L1** (5.2 mg, 0.033 mmol, 11 mol%), $CF_3CO_2Na$ (81.6 mg, 0.6 mmol, 2.0 equiv.) and $B_2pin_2$ (228.5 mg, 0.9 mmol, 3.0 equiv.). The flask was evacuated and refilled with argon, followed by the addition of $H_2O$ (16.2 μL, 0.9 mmol, 3.0 equiv.) and DMF (4 mL). The mixture was stirred at 80 °C for 10–20 h until the reaction was completed as monitored by TLC. The resultant solution was diluted with ethyl acetate, washed with HCl aqueous solution (1 M) and concentrated in vacuum. The mixture was detected by GC directly or after simple filtration in some cases to determine the *Z/E* ratio. The crude product was purified by chromatography on silica gel (300–400 mesh), eluted with petroleum ether with 0–20% of ethyl acetate to give alkene product. Careful column chromatography was able to partially deliver the major product in a pure form to provide precise NMR spectra of the major product. The overall isolated yield was calculated based on the combination of all parts. All compounds were characterized (see the Supplementary Information).

## Data availability

The data that support the findings of this study are available within the article, its Supplementary Information files. All data underlying the findings of this work are available from the corresponding author upon request. Supplementary Data 1 contains the data of the imaginary frequencies, free energies and coordinates of the optimized structures. Supplementary Data 2 contains the ${}^1H$, ${}^{19}F$, ${}^{13}C$ NMR spectra.

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

## Acknowledgements

Financial support from the National Natural Science Foundation of China (No. 21602015) (Y.L.) and Talent Development Fund of Jilin Province (Y.L.) are gratefully acknowledged. We thank Prof. Ruben Martin at Institute of Chemical Research of Catalonia (ICIQ) for his insightful discussion and generous help on the mechanistic studies.

## Author contributions

Y.W. developed the method and carried out most of the chemical reactions. Y.A., C.L. and J.Z. participated in the mechanistic studies. Z.L. conducted the DFT calculations. W.G., X.L., H.W. and Y.S.L. supported the synthesis of substrates. Y.L. designed and supervised the project. Y.L. and Y.W. wrote the manuscript.

## Competing interests

The authors declare no competing interests.
