## [Peer Review File · Nature Communications]

REVIEWER COMMENTS

Reviewer #1 (Remarks to the Author):

Recommendation: Publish with minor revisions

Summary and Key/New Developments:

The authors present a Ni-catalyzed stereodivergent semihydrogenation of alkynes with water. This semihydrogenation features green reagents and catalysts, where choice of base determines E- or Z-selectivity. Through a number of mechanistic studies including rate and EPR studies, the authors determined that the two products are formed via distinct pathways. A Ni(II) pathway is operative for the Z-alkene, while a Ni(I) pathway is operative for the E-alkene; in many stereodivergent syntheses of E- and Z-alkenes from alkynes, one isomer is usually derived via isomerization of the other product. The scopes of both reactions feature nearly 70 examples and tolerate a wide range of functional groups, including a number of diynes.

Questions and Comments:

- (1) Figure 1 features a wide array of data, and is a bit overwhelming. Some of the data could perhaps be left for the SI (specifically parts d and e). Organization of the figure is a bit confusing as well (e.g. part d is right above part g rather than e).
- (2) Substrate 1a was chosen as the model substrate, it would be interesting to know why.
- (3) Page 6, "The kinetic profiles of Z-selective semihydrogenation showed that 2a was generated by degrees...": it is unclear what "generated by degrees" means.

Grammatical/Graphical Errors

- (1) "TH" is used without being defined, all acronyms should be defined when first used.
- (2) Table 1 has the type: "dIsolated" instead of "disolated".
- (3) Page 6, "We postulate...metal center" is an incomplete sentence.

Reviewer #2 (Remarks to the Author):

The authors report a stereodivergent semihydrogenation of alkynes strategy by modulating Ni species at early stage with base and B2pin2 and using water as proton resource to synthesis Z- and

E-olefins with high selectivity. The scope of this strategy is great, and various internal and terminal triple bonds, enynes and diynes with different substituents were semi-hydrogenated to the target olefins in moderate to high yields. The study is interesting and, the obtained results are practical. Overall, I would be happy to recommend publication once the authors address the detailed comments below.

1. From Table 1 and 2, the selectivity ratio of main products to byproducts obtained (whatever Z/E in table 1 or E/Z in table 2) under optimal conditions is almost the same, Is there any reason for this?
2. For Figure 1b, can diborylated vinyl derivatives react with water under condition A or B?
3. EPR experiments in the absence of alkyne 1a should also be done to confirm that different Ni species are modulated and appeared at early stage.
4. What is the mechanism by which B2pin2 reduce $[\text{Ni}(\text{II})(\text{CF}_3\text{COO})_2\text{L}]$ to generate $[\text{Ni}(\text{O})\text{L}]$? how about the product distribution if adding reductant to afford Ni(I) species in condition A?
5. The authors are encouraged to isolate some intermediates and, in situ HRMS and NMR are also recommend to be conducted to prove the proposed mechanism. DFT calculations should be taken to deepen the insights of the mechanism, including the impact of base on the variation of Ni species and the process of isomerization from intermediate K to L.
6. How about the reactions if there is a hydroxyl or Bpin group on the aromatic ring?
7. There are some errors in the manuscript, which need to be carefully checked and corrected. For example, the full name of "TH" should be given when it was firstly referred; "1g" should be "1i" in line 90; "2a and 3a" should be "2g and 3g" in line 95; the letter "d" in the "dIsolated yield" in the footnote of Table 1 should be superscript; are the reaction conditions of "9 and 10" in Table 3 wrong?

Reviewer #3 (Remarks to the Author):

The present manuscript describes a protocol for a formal transfer hydrogenation of alkynes to alkenes. By choice of base (namely a benzoate vs. trifluoroacetate), the protocol delivers either E or Z alkenes from the corresponding alkynes. Generally, the field of stereoselective alkyne semihydrogenation has been quite active the last years, while many Z-selective protocols have been developed first, nowadays the E-selective methods are becoming more mainstay. This has ben summarized in several reviews (Chem. Rev. 2013, 113, 1313; ACS Catal. 2012, 2, 1773; Bull. Chem. Soc. Jpn. 2016, 89, 135) nonw of which has been cited here to guide the reader.

Several shortcomings can be found in this paper:

- The authors mention E/Z ratios as the main feature of their protocol. Not a single spectrum (the authors claim that ¹H NMR spectroscopy has been used to determine the E/Z ratios) has been presented in the SI that shows the ratio. Take for example compounds 2r and 3gg, only the pure compounds are presented. If there is no data for the actual ratios, as it stands right now, the most important claim of the paper is not supported by data. (The stereoisomers of the alkenes involved are very hard (if not impossible) to separate by column chromatography – how did the authors obtain pure spectra of all compounds?)
- The real challenge in this field is the stereoselective E-selective alkyne semihydrogenation of unactivated, non-conjugated alkenes. Aryl-substituted alkenes tend to work much better. Therefore, the present protocol, which almost exclusively focuses on (di)arylalkynes, does not represent a significant enlargement of the previous methods. In short, these are the “easy” substrates. Dialkyl ones are hard.
- The title of the manuscript is highly misleading, if not wrong. The present protocol is a borylation/protodeborylation, with a lot of good will it might be in total a transfer semihydrogenation, but it is certainly no semihydrogenation itself, as the authors claim. This key change makes the present results significantly less important. Alkyne semireductions with stoichiometric reducing agents other than H₂ are well-known (see the reviews above, also the Birch method) and the real challenge in this field are processes just based on H₂. It should also be mentioned that the borylation/protodeborylation has been reported before, see for example here (there are more examples): Chem. Commun., 2019,55, 6922 So, therefore, the present approach is not new.
- Along these lines, the citation of literature to guide the reader is not acceptable. No key alkyne semihydrogenation with Ni catalysts are cited (see, for example: Chem Eur J 2020, 26, 1597; ChemSusChem 2019, 12, 3363, and references therein). This is important for the readers to judge the key challenges in this field. Furthermore, a protocol based on Zn as stoichiometric reducing agent that is very similar to this work is not cited (DOI: 10.1002/ajoc.202000716), which makes the present work lose much of its novelty.
- The authors claim that the present protocol actually delivers two distinct catalytic species and support this claim by EPR spectroscopy. There is a general misconception made by the authors here: Just because something is observable, does not mean it actually is a catalytically active species. Actually, the opposite makes much more sense. Catalytically active compounds should be very hard to detect (especially on the EPR timescale) as they have a limited lifetime, off-cycle intermediates and “dead-ends” have a longer lifetime, but by definition do not lie on the catalytic cycle. Therefore, this referee highly questions the validity of the claim the authors make based on the EPR spectroscopy.
- The authors claim that the E-selective protocol is not based on an Z-E-isomerization process. In Figure 1c, the authors actually show the formation of a small amount of Z-alkene, simultaneous to the formation of the E-product. This is typical of a fast isomerization step. Therefore, yet another main claim of the paper remains highly questionable.

- The fact that the starting alkyne is involved in Ni-catalyzed alkyne semihydrogenation has been reported before and is not new, the authors should refer to this (Chem Eur J 2020, 26, 1597; ChemSusChem 2019, 12, 3363).

- There are several typing and factual mistakes, making the overall manuscript hard to read.

On the whole, the present manuscript fails to support its main claims (unclearness about semihydrogenation / transfer semihydrogenation – the present borylation/deborylation approach is not new, a very similar paper has recently been published based on Zn as reducing agent) as outlined above. Furthermore, the E/Z ratios cannot be backed up by data. Literature citation is not acceptable. The main claim of a non-isomerization pathway cannot be hold up in view of the data. Therefore, the present manuscript does not hold quality requirements for publication. This referee suggests the rejection of the manuscript in the present form.

Dear Sir/Madam,

On behalf of my co-authors, we thank you very much for your positive and constructive comments and suggestions on our manuscript entitled “*Modulation of Metal Species as Early Control Point for Ni-catalyzed Stereodivergent Semihydrogenation of Alkynes with Water*”. We have studied the comments carefully and made corresponding corrections. Revised parts are yellow marked in the paper. A point by point response to the comments are listed as follows.

Reviewer 1:

Recommendation: Publish with minor revisions

Summary and Key/New Developments:

The authors present a Ni-catalyzed stereodivergent semihydrogenation of alkynes with water. This semihydrogenation features green reagents and catalysts, where choice of base determines E- or Z-selectivity. Through a number of mechanistic studies including rate and EPR studies, the authors determined that the two products are formed via distinct pathways. A Ni(II) pathway is operative for the Z-alkene, while a Ni(I) pathway is operative for the E-alkene; in many stereodivergent syntheses of E- and Z-alkenes from alkynes, one isomer is usually derived via isomerization of the other product. The scopes of both reactions feature nearly 70 examples and tolerate a wide range of functional groups, including a number of diynes

Questions and Comments:

(1) Figure 1 features a wide array of data, and is a bit overwhelming. Some of the data could perhaps be left for the SI (specifically parts d and e). Organization of the figure is a bit confusing as well (e.g. part d is right above part g rather than e).

Response: Thanks very much for the kind suggestion. Figure 1d and 1e in the original manuscript have been moved to Supplementary Information. The figure in the revised manuscript has been organized accordingly.

(2) Substrate 1a was chosen as the model substrate, it would be interesting to know why.

Response: The reasons for choosing Substrate **1a** are as follows: (1) The reactivity of aryl alkyne substrates is not specific, and it can better represent the reducibility of semi-hydrogenation of alkynes. (2) The reaction system can be analyzed unambiguously with characteristic NMR peaks of methyl.

(3) Page 6, “The kinetic profiles of Z-selective semihydrogenation showed that 2a was generated by degrees...”: it is unclear what “generated by degrees” means.

Response: We are sorry for the unclear expression. We wanted to express that the amount of **2a** increases gradually as the reaction proceeds. The sentence has been revised to “The kinetic profile of Z-selective semihydrogenation system showed that the concentration of **2a** increased gradually throughout the reaction period, staying closely aligned with the conversion of **1a**.”

Grammatical/Graphical Errors

(1) “TH” is used without being defined, all acronyms should be defined when first used.

(2) Table 1 has the type: “dIsolated” instead of “dIsolated”.

(3) Page 6, “We postulate...metal center” is an incomplete sentence.

Response: We are awfully sorry for our carelessness. We have checked through the whole manuscript and made necessary corrections. The mentioned parts by the reviewer have been corrected as follows:

(1)

systems to realize a *Z* to *E* isomerization process at the late stage (Scheme 1, above). For instance, Moran, *et al* showed that Ni-catalyzed **transfer hydrogenation (TH)** of alkynes with HCO₂H selectively afforded *Z*-olefins, which isomerized to *E*-isomers by adding triphos ligand.⁸ Another catalyst-modulated system was disclosed by Liu and coworkers in 2016, in

(2)

^aReactions were performed with **1a** (0.15 mmol), NiBr₂ (10 mol%), **L1** (22 mol%), base (2.0 equiv.), B₂Pin₂ (3.0 equiv.), H₂O (3.0 equiv.), DMF (2 mL), 80 °C, 10 h; ^b5 mol% of NiBr₂, 11 mol% of **L1**; ^cDetermined by crude ¹H NMR; ^disolated yield; ^eonly *Z*-alkene product **2a**

(3) This sentence was corrected as “We postulate that the small amount of *E*-alkene in this system was generated from isomerization of the *Z*-product, which was suppressed in the initial period due to competitive coordination of alkyne **1a** with the metal center.”

Reviewer #2 (Remarks to the Author):

The authors report a stereodivergent semihydrogenation of alkynes strategy by modulating Ni species at early stage with base and B₂pin₂ and using water as proton resource to synthesis Z- and E-olefins with high selectivity. The scope of this strategy is great, and various internal and terminal triple bonds, enynes and diynes with different substituents were semi-hydrogenated to the target olefins in moderate to high yields. The study is interesting and, the obtained results are practical. Overall, I would be happy to recommend publication once the authors address the detailed comments below.

1. From Table 1 and 2, the selectivity ratio of main products to byproducts obtained (whatever Z/E in table 1 or E/Z in table 2) under optimal conditions is almost the same, Is there any reason for this?

Response: Thanks very much for the review. We regard the same ratio for the hydrogenation of **1a** in both *Z*- and *E*-selective conditions as a coincidence, since most other alkynes delivered the olefinic products in different ratios.

2. For Figure 1b, can diborylated vinyl derivatives react with water under condition A or B?

Response: Thanks for the kind suggestion. The corresponding experiments have been carried out. As shown below, diborylated vinyl derivatives didn't react with water in either conditions, leaving most starting materials recovered. The results have been included in the revised manuscript.

Condition A: NiBr₂ (10 mol%), **L3** (22 mol%), PhCO₂Na (1.0 equiv.), B₂Pin₂ (3.0 equiv.), H₂O (3.0 equiv.), DMF, 80 °C in Ar atmosphere;
Condition B: NiBr₂ (10 mol%), **L1** (22 mol%), CF₃CO₂Na (2.0 equiv.), B₂Pin₂ (3.0 equiv.), H₂O (3.0 equiv.), DMF, 80 °C in Ar atmosphere;

3. EPR experiments in the absence of alkyne **1a** should also be done to confirm that different Ni species are modulated and appeared at early stage.

Response: Thanks very much for the professional suggestion. EPR experiments in the absence of alkyne **1a** have been carried out accordingly, which are shown in the figure below. In consistent with the studies in the presence of starting materials, no signal was observed when PhCO₂Na was used, while a strong EPR signal was showed up with CF₃CO₂Na as base, demonstrating that different Ni species are modulated and appeared at early stage without the participation of alkynes (left). In addition, we also analyze the mixture of NiBr₂ and Ni(cod)₂, which would generate Ni(I) species in situ (e.g. *Nat. Catal.* 2021, 4, 124-133; *Angew. Chem. Int. Ed.* 2017, 56, 13431-13435; *J. Am. Chem. Soc.* 2017, 139, 922-936). The similar signal compared with the system using CF₃CO₂Na further proved our proposal (right). The results have been added in the revised paper along with necessary discussions.

4. What is the mechanism by which B₂pin₂ reduce [Ni(II)(CF₃COO)₂L] to generate [Ni(0)L]? how about the product distribution if adding reductant to afford Ni(I) species in condition A?

Response: Thanks very much for the guiding question, which really paves our way to further mechanistic understanding. Regarding the mechanism on the reduction of [Ni(II)(CF₃COO)₂L] to [Ni(0)L] by B₂pin₂, density functional theory (DFT) calculation was studied. The reaction free-energy profile is shown below. Firstly, LNiBr₂ **A** reacted with CF₃CO₂Na to give Ni(II) species **G**, which is exergonic by 48.4 kcal/mol. One- or two-ligand exchange with B₂Pin₂ producing X-[NiL]-BPin (**H** or **P**) is endothermic with 13.4 and 24.4 kcal/mol, respectively. The notably higher ΔG value of two-ligand exchange makes it more reluctant to happen. For the one-ligand exchange pathway, subsequent reductive elimination and comproportionation process are exergonic, leading to Ni(I) species **J** and **K** to initiate the E-selective catalytic cycle.

Base: CF₃CO₂Na

The experiments with the addition of reductants were also carried out. The reactivity in condition A was suppressed using Zn or Mn instead of B₂Pin₂. Besides, the *Z/E* ratios dropped when reductant (Mn or Zn) was added into condition A, which might due to the generation and competitive act of Ni(I) species, proving that the presence of Ni(I) species is indeed conducive to the formation of *E*-selective products. Furthermore, in situ formed Ni(I) species by mixing NiBr₂ and Ni(cod)₂ resulted in olefin **3a** with *E*-configuration as major product (40% yield, 18/82 *Z/E*). Finally, it is noteworthy that the alkynes could not undergo any transformation in this hydrogenation system without base (ESI- Supplementary Table 3, entry 24), which is another evidence that Ni(I) species was generated by the aid of CF₃CO₂Na and tended to produce *E*-olefins. The above DFT calculations and control experiments have been added in Figure 2.

5. The authors are encouraged to isolate some intermediates and, in situ HRMS and NMR are also recommend to be conducted to prove the proposed mechanism. DFT calculations should be taken

to deepen the insights of the mechanism, including the impact of base on the variation of Ni species and the process of isomerization from intermediate K to L.

Response: Thanks for the kind suggestion. In situ NMR was monitored at 80 °C to find some supporting evidence. The Z-selective system with PhCO₂Na as base displayed a new peak at the chemical shift of 4.69 ppm, which still remains in the absence of water and alkyne **1a**. This new peak might be assigned to the methyl of Bpin in the [PhCOONi(II)BpinL] **C** species, since the NMR chemical shifts of protons in the metal complex might move to the low-field area (ACS Catal. 2020, 10, 2117-2123 ESI-page S15). Meanwhile, a new peak at 3.33 ppm in the E-selective system using CF₃CO₂Na as base was observed, which did not disappear in the absence of water and **1a**. This new peak may be the signal of [LNi(I)Bpin] **J** species in the E-selective catalytic cycle. A similar signal (3.27 ppm) was found in the in-situ NMR of the mixture of NiBr₂, Ni(cod)₂ and B₂Pin₂, further supporting our speculation. A faint peak in the chemical shift of 4.69 ppm was observed in the close-up of 4 ppm to 6 ppm (line 4 and 5), which is probably the hydrogen signal of [CF₃COONi(II)BpinL] **H** species corresponded with [PhCOONi(II)BpinL] **C**. Further NMR and HRMS studies didn't provide any valid data at this stage. We also tried to isolate possible intermediates in the catalytic hydrogenation system, including metal intermediates and boron species, but ended in failure. We will keep focusing on the mechanistic studies for in-depth insight.

Density functional theory (DFT) calculations were carried out to investigate the remarkably different impact of PhCO_2Na and $\text{CF}_3\text{CO}_2\text{Na}$ on Ni species. The reaction free-energy profiles are shown below. The Ni(II) precursor **A** reacted with PhCO_2Na to afford nickel benzoate **B**, which is exergonic by 28.9 kcal/mol (left). One- or two-ligand exchange of **B** with B_2pin_2 towards X-Ni(II)-BPin (**C** or **P**) is favored thermodynamically. Subsequent reductive elimination and comproportionation process could deliver thermodynamically more stable Ni(I) species **J** or **Q**. However, activation of **C** with H_2O molecule en route to Ni(II)-H species **D** is more favored than the above process, supporting our mechanistic investigations and conjecture that the Z-selective catalytic cycle was initiated by $[\text{PhCOO-Ni(II)L-BPin}]$ species. The right part in the Figure below shows the DFT computation results using $\text{CF}_3\text{CO}_2\text{Na}$ as base. Firstly, LNiBr_2 **A** reacted with $\text{CF}_3\text{CO}_2\text{Na}$ to give Ni(II) species **G**, which is exergonic by 48.4 kcal/mol. One- or two-ligand exchange with B_2Pin_2 producing X-[NiL]-BPin (**H** or **P**) is endothermic with 13.4 and 24.4 kcal/mol, respectively. The notably higher ΔG value of two-ligand exchange makes it more reluctant to happen. For the one-ligand exchange pathway, subsequent reductive elimination and comproportionation process are exergonic, leading to Ni(I) species **J** and **K** to initiate the E-selective catalytic cycle.

DFT calculations show that the isomerization from intermediate **K** to **L** (**M** to **N** in revised manuscript) was found to proceed via a three-membered ring transition state (as show below), with an activation barrier of 26.7 kcal/mol. The *trans* isomer **N** is 2.3 kcal/mol more stable than **M**.

The above NMR analysis and DFT calculations have been added in Supplementary Figure 2, Figure 2f and 2g. Thanks again for the reviewer's professional suggestions.

6. How about the reactions if there is a hydroxyl or Bpin group on the aromatic ring?

Response: Thanks for the kind suggestion. The two substrates with hydroxyl and Bpin groups have been synthesized and tested. Arylalkyne **1u** bearing hydroxyl group provided the alkene product with 70/30 *Z/E* ratio. The relatively lower selectivity might be caused by isomerization of *Z*-olefin **2u**, since the *Z/E* ratio of the olefins decreased from 86/14 to 53/47 by heating in DMF at 80 °C for 9 h. It is within expectation that the reaction of **1u** under *E*-selective conditions delivered **3u** in excellent yield and full selectivity (95% yield, >99/1 *Z/E*). Bpin-substituted alkyne **1v** underwent *Z*- and *E*-selective hydrogenation smoothly, furnishing **2v** in 78% yield and >99/1 *Z/E* as well as **3v** in 64% yield and 79/21 *E/Z*. The relatively lower selectivity in the latter case might be caused by competitive interaction of the Bpin group with active species in the reaction system. The above results have been included in the revised manuscript.

7. There are some errors in the manuscript, which need to be carefully checked and corrected. For example, the full name of “TH” should be given when it was firstly referred; “1g” should be “1i” in line 90; “2a and 3a” should be “2g and 3g” in line 95; the letter “d” in the “dIsolated yield” in the footnote of Table 1 should be superscript; are the reaction conditions of “9 and 10” in Table 3 wrong?

Response: Thanks very much for the careful check. We have checked through the manuscript. The mentioned parts have been revised as follows:

(1)

systems to realize a *Z*-to-*E* isomerization process at the late stage (Scheme 1, above). For instance, Moran, *et al.* showed that Ni-catalyzed **transfer hydrogenation (TH)** of alkynes with HCO₂H selectively afforded *Z*-olefins, which isomerized to *E*-isomers by adding triphos ligand.⁸ Another catalyst-modulated system was disclosed by Liu and coworkers in 2016, in

(2)

conditions (equations (1) and (2)). Similar results were also observed for unsymmetric alkynes **1bb** and **1i**, with the former leading to even higher deuterations (equations (7) and (8)). In contrast, there was no sign of deuteration on the products

(3)

consequent hydrogenation was not involved in the catalytic pathway. Control experimental studies of vinylboron reagents **4** and **5** were performed under the standard reaction conditions.^{39,40} Olefin products **2g** and **3g** were not detected (Figure 1b, equations (9) and (10)). **Diborylated vinyl derivatives 4' and 5'** were also untouched in both conditions, leaving most

(4)

^aReactions were performed with **1a** (0.15 mmol), NiBr₂ (10 mol%), L1 (22 mol%), base (2.0 equiv.), B₂Pin₂ (3.0 equiv.), H₂O (3.0 equiv.), DMF (2 mL), 80 °C, 10 h; ^b5 mol% of NiBr₂, 11 mol% of L1; ^cDetermined by crude ¹H NMR; ^dIsolated yield; ^eonly *Z*-alkene product **2a**

(5) The reaction conditions of “9 and 10” in Table 3 are correct. The exact reason for the “unexpected” outcomes might be related to the stability of the major products.

Reviewer #3 (Remarks to the Author):

The present manuscript describes a protocol for a formal transfer hydrogenation of alkynes to alkenes. By choice of base (namely a benzoate vs. trifluoroacetate), the protocol delivers either E

or *Z* alkenes from the corresponding alkynes. Generally, the field of stereoselective alkyne semihydrogenation has been quite active the last years, while many *Z*-selective protocols have been developed first, nowadays the *E*-selective methods are becoming more mainstay. This has been summarized in several reviews (*Chem. Rev.* 2013, 113, 1313; *ACS Catal.* 2012, 2, 1773; *Bull. Chem. Soc. Jpn.* 2016, 89, 135) none of which has been cited here to guide the reader.

Response: Thanks for the professional remarks. Indeed, semihydrogenation of alkynes have attracted remarkable interests in recent years. Many precedents have been disclosed for selective transformation, including some elegant advances for controllable delivery of both isomers (e.g. *Nature Catalysis*, 2019, 2, 529-536; *J. Am. Chem. Soc.* 2016, 138, 8588-8594; *ACS Catal.* 2021, 11, 13696-13705). Considering the importance of this field, we think it is still desirable to develop novel and convenient alternatives in both academic and practical aspects.

Mechanistically, *E*-olefins are generally derived from the isomerization of the corresponding *Z*-alkenes. Therefore, it normally requires employment of different catalysts or careful supervise of the reaction conditions to achieve high selectivity. In contrast, we disclosed an unprecedented strategy, in which judicious choice of bases resulted in different nickel species starting from the same catalyst precursor, paving the way to both isomers in totally distinct catalytic cycles. Besides the novelty in hydrogenation chemistry, we believe our system deserves notice for better understanding and further developments of nickel catalysts. We will further focus on extension of the protocol and also exploration of the in-depth mechanism.

The reaction system is also featured by involvement of nonprecious catalysts and ligands with nontoxic additives. Water was used as the single hydrogen donor. The robustness of the protocol was showcased with nearly 70 examples, including internal and terminal alkynes, enynes and diynes. The reviewer's kind suggestions also guided us to a more systematic and convictive study. We hope the revised submission would meet with the criteria of the reviewers and also the journal. The references mentioned by the reviewer have been added into the manuscript. Thanks again for the kind suggestion.

Several shortcomings can be found in this paper:

- The authors mention E/Z ratios as the main feature of their protocol. Not a single spectrum (the authors claim that ¹H NMR spectroscopy has been used to determine the E/Z ratios) has been presented in the SI that shows the ratio. Take for example compounds 2r and 3gg, only the pure compounds are presented. If there is no data for the actual ratios, as it stands right now, the most important claim of the paper is not supported by data. (The stereoisomers of the alkenes involved are very hard (if not impossible) to separate by column chromatography – how did the authors obtain pure spectra of all compounds?)

Response: Thanks for the suggestion. NMR analysis of the crude mixture was employed to determine the *E/Z* ratios, which was also widely used in previous reports for semi-hydrogenation of alkynes (*J. Am. Chem. Soc.* 2015, 137, 14598-14601, *ACS Catal.* 2017, 7, 8296-8303, *Org. Lett.* 2019, 21, 1412-1416). For some special cases which is difficult to judge from the original mixture, simple filtration or separation of the major isomer were needed to make a rigorous report. Representative crude ¹H NMR spectra have been added into ESI, including **2r** and **3gg** (**3hh** in revised manuscript) mentioned by the reviewer.

Indeed, the stereoisomers of the alkenes are really difficult to separate. Fortunately, by column

chromatography (300-400 mesh silica gel) using test-tubes to carefully catch the eluant, we were able to obtain the products in pure.

- *The real challenge in this field is the stereoselective E-selective alkyne semihydrogenation of unactivated, non-conjugated alkenes. Aryl-substituted alkenes tend to work much better. Therefore, the present protocol, which almost exclusively focuses on (di)aryllalkynes, does not represent a significant enlargement of the previous methods. In short, these are the “easy” substrates. Dialkyl ones are hard.*

Response: Indeed, *E*-selective semihydrogenation of non-conjugated alkynes is challenging (e.g. *Nature Catalysis*, 2019, 2, 529-536; *J. Am. Chem. Soc.* 2016, 138, 8588-8594; *ACS Catal.* 2021, 11, 13696-13705). In our study, dialkyl alkyne **1ii** delivered **3ii** in only 52/48 *E/Z* ratio under the *E*-selective conditions, although the *Z*-isomer **2ii** was accessed with almost a single configuration under the other condition. The results have been included in the revised manuscript. Although imperfect, we think the novelty, simplicity of our protocol as well as the robustness showcased by the almost 70 examples in the scope have demonstrated the advancement of the system. We will focus on further investigations including the extension of the hydrogenation strategy toward other type of substrates in the future.

- *The title of the manuscript is highly misleading, if not wrong. The present protocol is a borylation/protodeborylation, with a lot of good will it might be in total a transfer semihydrogenation, but it is certainly no semihydrogenation itself, as the authors claim. This key change makes the present results significantly less important. Alkyne semireductions with stoichiometric reducing agents other than H₂ are well-known (see the reviews above, also the Birch method) and the real challenge in this field are processes just based on H₂. It should also be mentioned that the borylation/protodeborylation has been reported before, see for example here (there are more examples): *Chem. Commun.*, 2019,55, 6922 So, therefore, the present approach is not new.*

Response: Borylation/protodeborylation process has indeed been well established as pointed out by the reviewer. However, the possibility has been excluded since vinylboron derivatives did not lead to the olefin products under the standard hydrogenation conditions (Scheme 1b, equations (9) and (10)). For the referee's full consideration, the reactivity of diborylated olefins under the conditions were also tested, with most starting materials untouched (Scheme 1b, equations (11) and (12)). Based on all the experimental results and related reports (*J. Am. Chem. Soc.* 2016, 138, 6107-6110; *Angew. Chem. Int. Ed.* 2017, 56, 3987-3991), we believe that the role of B₂Pin₂ in this system is to activate water as well as to interact with bases to generate different catalyst species. We appreciate the reviewer's remarks, which guided us to more detailed mechanistic studies. The mentioned reference has been cited in the revised submission (ref.40).

- *Along these lines, the citation of literature to guide the reader is not acceptable. No key alkyne semihydrogenation with Ni catalysts are cited (see, for example: *Chem Eur J* 2020, 26, 1597; *ChemSusChem* 2019, 12, 3363, and references therein). This is important for the readers to judge*

the key challenges in this field. Furthermore, a protocol based on Zn as stoichiometric reducing agent that is very similar to this work is not cited (DOI: 10.1002/ajoc.202000716), which makes the present work lose much of its novelty.

Response: Thanks for the remarks. Although Ni-catalyzed alkyne semihydrogenation has been developed in some pioneer reports, the chemistry in our system is distinct from them. As verified in our work, simple modulation of the bases serves as a key control point, which lead to totally different nickel species. Olefins with different configurations were delivered subsequently in separate catalytic cycles. The role of B₂Pin₂ is not only to activate water, but also to facilitate nickel species with different valence, which is distinct from Zn participated reactions. Based on the reviewers' kind suggestions, control experiments with reductants such as Mn and Zn instead of B₂Pin₂ were conducted, which further demonstrated the unique effect of the boronic agents as the former two reductants failed to provide the desired olefins in our reaction system without B₂Pin₂ (Figure 2c). The simplicity of the reaction conditions employing very ordinary reagents including the nickel precursors, ligands, bases and also water as the hydrogen source makes our protocol "seemingly featureless", but we think the in-depth chemistry is meaningful, and simple reactions involving common factors are always welcomed during the exploration of new methodologies. The revised paper provides more evidence to support our standpoint, which we hope could be convincing for the reviewer. The mentioned references have been cited in the revised manuscript (ref. 28-30). Thanks again for the comments.

- The authors claim that the present protocol actually delivers two distinct catalytic species and support this claim by EPR spectroscopy. There is a general misconception made by the authors here: Just because something is observable, does not mean it actually is a catalytically active species. Actually, the opposite makes much more sense. Catalytically active compounds should be very hard to detect (especially on the EPR timescale) as they have a limited lifetime, off-cycle intermediates and "dead-ends" have a longer lifetime, but by definition do not lie on the catalytic cycle. Therefore, this referee highly questions the validity of the claim the authors make based on the EPR spectroscopy.

Response: Thanks for the professional remarks. Although all the experimental and analytic results point to different nickel species, further experimental supports are indeed awaited to validate the proposed mechanism. According to the reviewers' kind suggestions, more detailed studies were carried out in order to improve the credibility of our proposal. For example, extra Zn and Mn were added into the Z-selective hydrogenation conditions, which would afford Ni(I) species in the system. As a result, the ratio of the E-olefin was promoted, supporting our hypothesis that Ni(I) species would be the active species in the E-selective catalytic cycle. In consistent with the above results, in situ formed Ni(I) species by mixing NiBr₂ and Ni(cod)₂ also furnished E-olefin **3a** as the major product. Extra EPR analysts based on the reviewers' suggestions were also in consistent with previous results. Finally, DFT calculations were appealed to support the mechanism. All the above investigations have been included in the revised manuscript. More detailed studies will be conducted in our lab for better understanding of the reaction system.

- The authors claim that the E-selective protocol is not based on an Z-E-isomerization process. In Figure 1c, the authors actually show the formation of a small amount of Z-alkene, simultaneous to

the formation of the *E*-product. This is typical of a fast isomerization step. Therefore, yet another main claim of the paper remains highly questionable.

Response: The formation of the small amount of *Z*-alkene seems independent from the generation of the *E*-product. As shown in previous reports (*Nature Catalysis*, 2019, 2, 529-536; *J. Am. Chem. Soc.* 2016, 138, 8588-8594; *Org. Lett.* 2019, 21, 1412-1416), different distribution of the starting material and products should be observed for a *Z/E* isomerization process. Control experiments referring to existing references (*Nature Catalysis*, 2019, 2, 529-536; *J. Am. Chem. Soc.* 2016, 138, 8588-8594; *ACS Catal.* 2021, 11, 13696-13705) were also conducted for further support. Specifically, when *Z*-alkene **2a** was put in the standard condition B, only 3% of *E*-alkene was detected, demonstrating the reluctance of the *Z/E* isomerization process. The results are presented in the revised submission (Supplementary Scheme 1, equation (1)).

- The fact that the starting alkyne is involved in Ni-catalyzed alkyne semihydrogenation has been reported before and is not new, the authors should refer to this (*Chem Eur J* 2020, 26, 1597; *ChemSusChem* 2019, 12, 3363).

Response: Despite recent reports on Ni-catalyzed alkyne semihydrogenation have been presented, simple alternative protocols still deserve exploration considering the importance of such transformation. Catalytic reactions involving novel mechanistic pathways would be more meaningful. Specifically, the two reports referred by the referee and also most previous ones employed H₂ as the hydrogen source. Although being the most atomic economic, alternative hydrogen sources, especially water possess unique advantages such as safety, availability, operability and so on. For other reaction factors, all catalysts and additives are commercially available and inexpensive, shunning the preparation of complex reagents, making the system simpler and more efficient. Mechanistically, an unprecedented reaction pathway has been disclosed, in which bases play a vital role to modulate the active Ni species from the same catalyst precursor, allowing for the realization of stereodivergent transformation in a unique manner. We hope the novelty of the system would feature our work. We believe the studies presented by us would lead to further explorations on alkyne transformations and also nickel catalysts. We hope the revised manuscript would get the referee's vote.

- There are several typing and factual mistakes, making the overall manuscript hard to read.

Response: We are awfully sorry for our careless. We have checked through the whole manuscript and made necessary corrections. We hope the revised paper would be more readable.

On the whole, the present manuscript fails to support its main claims (unclearness about semihydrogenation / transfer semihydrogenation – the present borylation/deborylation approach is not new, a very similar paper has recently been published based on Zn as reducing agent) as outlined above. Furthermore, the E/Z ratios cannot be backed up by data. Literature citation is not acceptable. The main claim of a non-isomerization pathway cannot be hold up in view of the data. Therefore, the present manuscript does not hold quality requirements for publication. This referee suggests the rejection of the manuscript in the present form.

Response: We are sorry that our submission didn't meet with the criteria of the referee. We have tried to answer all the questions listed above and conducted more investigations including functional group tolerance, mechanistic experiments, DFT calculations, among others to further support our viewpoint, which we hope to convince the referee on our proposal. We are thankful for the professional comments, which help us on further revisions to make a better work.

REVIEWER COMMENTS

Reviewer #1 (Remarks to the Author):

The authors have provided a satisfactory response to my initial review and I am supportive of publication.

Reviewer #2 (Remarks to the Author):

< In private comments to the Editor, the Reviewer said that their concerns were addressed, and that they supported publication at this stage. >

Reviewer #3 (Remarks to the Author):

The present revised version of the manuscript takes up many of the referee's comments and the authors have made a substantial effort to follow the suggestions of the referee's to improve the paper. It is indeed a curious study that the authors present, as they now show substantial evidence that indeed their alkyne transfer semihydrogenation operates via two different pathways to either the Z or the E-product. As this stands, the work is of general interest to the community and certainly makes a nice contribution to the field which is certainly worthy of publication. However, claims have to be backed up by proper data, which in the eyes of this referee is still not the case here.

- In my earlier report, I raised the point that the key claim of this paper, namely the high E or Z selectivity in the transfer semihydrogenation is not backed up by proper data. The authors have responded to this by adding some crude spectra of the compounds I mentioned as examples (sic!) in my last report. So, for very few of the compounds now the ¹H NMR spectra are given. This is not the case for compounds 2c, 2e, 2f, 2h, 2k, 2l (here it should have been easy employing ¹⁹F NMR), 2m, 2q; resolution is not acceptable for 2d, 2j, 2o, 2p (here, the authors also miscalculated the ratios); for 2g a crude spectrum is given but it is completely unclear where the E/Z ratio comes from as no peaks are picked. This is where I stop.

I repeat my criticism that the main claim to high stereoselectivity is not backed up by proper data. Why is there no GC data? This would have been the standard way to report ratios of diastereomers. Furthermore, many of the high ratios as claimed by the authors have values which are outside of the instrumental preciseness of an NMR spectrometer. Again, GC data would have helped to solve this.

- Based on the spectral evidence as outlined above and extensively in my first report, it is not clear how the authors obtain pure spectra of the final alkenes. In their response to my first report, the

authors have added a sentence to the SI that the diastereomers were separated by “simple filtration” (the authors do not give the filtering agent here). It is simply not possible to remove an unwanted E/Z-isomer by filtration unless special filtering agents such as Ag-coated silicagel has been employed. Furthermore, based on the experience in my group on catalytic alkyne (transfer) semihydrogenation, I claim based on my year-long experience that for most of the substrates presented here a separation via column chromatography is simply not possible. Therefore, it remains unclear how the authors come up with clean spectra such as the ones presented.

- The authors seemingly disregard my earlier comment on the difference between transfer semihydrogenation and semihydrogenation and keep the title of their manuscript the same.

In the eyes of this referee, there is certainly nice chemistry presented in this paper, as outlined above. However, when the key claims cannot be verified nor falsified, the paper does not hold to scientific standards. I therefore suggest the rejection of the manuscript.

This work presents a Ni-catalyzed stereodivergent semihydrogenation of acetylenes with water. The strategy features use of cheap catalysts and nontoxic reagents, tolerance to a wide range of alkynyl substrates, and moderate to high yields of olefinic products. The authors proposed base modulates the valence state of active nickel species at an early stage, and consequently leads to *Z*- and *E*-isomers in disparate catalytic pathways: the in situ formed Ni(II) species delivers *Z*-alkenes, while the Ni(I) species affords *E*-alkenes as final products.

I think the study is of significance in both academic and practical perspectives, if the authors can offer more adequate computational data to justify their proposed mechanism.

Since no key intermediate could be isolated, as the authors stated, energy data based on calculation of key stationary points is a good alternate. However, from the view of theoretical study, it is not rigorous enough to identify the dominant reaction pathway only by comparison of the relative energies of forked intermediates or products, some key transition states (TSs) should be located.

1. Based on the materials available, I think the authors want to address two important points about their catalytic cycles in Fig 2(e), that is, (1) no Ni(I) species (**J** or **J+Q** in Fig 2(f)) can be formed when PhCOONa is used, while Ni(I) species (**J** and **K**) is possible in the CF₃COONa case; (2) when PhCOONa is used, the intermediate **E** does not isomerize to **E'** (*E*-isomer of **E**), while intermediate **M** can isomerize to **N** in the CF₃COONa case. To rationalize these two points theoretically, the authors are suggested to conduct further calculations as follows:

(1) Locate TSs corresponding to **C**→**D**, **C**→**I+C**(or **J+Q**) and **B**→**C**, **B**→**P** processes in the left part of Fig 2(f), and exclude the possibility of **J** and **J+Q**'s formation when taking PhCOONa as base, through comparison of corresponding activation free energy barriers.

(2) Locate TSs corresponding to **H**→**I** and **H+I**→**J+K** processes in the right part of Fig 2(e), and demonstrate the possibility of **J+K**'s formation when taking CF₃COONa as base, under an experimental temperature of 80°C.

(3) Locate transition state (TS) corresponding to **H**→**D'**(replace PhCOO- group with CF₃COO- group in **D**) process in the right part of Fig 2(e), and exclude the possibility that intermediate **H** participates the catalytic cycle directly through transformation into intermediate **D'** when CF₃COONa is used, through comparison of corresponding activation free energy barriers.

(4) Locate TS corresponding to **E**→**E'** process in the left part of Fig 2(f), and exclude the possibility of **E**→**E'** transformation when taking PhCOONa as base, through comparison of corresponding activation free energy barriers.

2. On page 15, in the SI file, the author stated, "The intrinsic reaction coordinate (IRC) calculations was also performed to verify the connectivity of the transition state and the energy minima". However, I can't find corresponding IRC results there or in the text. It's suggested to be added at least into the SI file, in the right place.

3. On page 6, in the rebuttal file, the author stated, "However, activation of **C** with H₂O molecule en route to Ni(II)-H species **D** is more favored than the above process, supporting our mechanistic investigations and conjecture that the Z-selective catalytic cycle was initiated by [PhCOO-Ni(II)L-BPin] species". I don't agree well with this statement. As shown in Fig 2(f) in the text, although **D** is relatively more stable when comparing with **J+Q**, this is not sufficient to rule out the possibility of **J+Q**'s formation from **C**, because both the formation of **D** and **J+Q** from **C** are obviously exothermic processes. Considering **J+Q** is ~40 kcal/mol lower than **C** in the potential energy surfaces, the reverse pathway (from **J+Q** to **C**) is nearly impossible under an experimental temperature of 80°C, that is, once **J+Q** is formed, it can no longer be transformed to **D**. So, in the absence of activation free energy barriers corresponding to transformations from **C** to **D** and to **I+C** (or **J+Q**), it is not rigorous enough to identify **D** as the only product (intermediate). In addition, **P** is slightly more stable than **C**, possible conversion of intermediate **B** to **J** via **P** should be excluded, too.

4. If the authors submit a new version, it is recommended to add different symbols onto the potential energy profiles in Fig 2(f), so that readers can distinguish different reaction pathways easily in the printed version.

If the authors can meet all points listed above, I am happy to recommend publication on *Nature Communications*.

Dear Sir/Madam,

On behalf of my co-authors, we thank you very much for your positive and constructive comments and suggestions on our manuscript entitled "Modulation of Metal Species as Early Control Point for Ni-catalyzed Stereodivergent Transfer Semihydrogenation of Alkynes with Water". We are also grateful for giving us extra time to finish the revision. We have studied the comments carefully and made corresponding corrections. Revised parts are yellow marked in the paper. A point by point response to the comments are listed as follows.

Reviewer #1:

The authors have provided a satisfactory response to my initial review and I am supportive of publication.

Reviewer #2:

In private comments to the Editor, the Reviewer said that their concerns were addressed, and that they supported publication at this stage.

Response: Thanks for the reviewers for their affirmation. The guiding comments from them helped us a lot to improve the manuscript.

Reviewer #3:

The present revised version of the manuscript takes up many of the referee's comments and the authors have made a substantial effort to follow the suggestions of the referee's to improve the paper. It is indeed a curious study that the authors present, as they now show substantial evidence that indeed their alkyne transfer semihydrogenation operates via two different pathways to either the Z or the E-product. As this stands, the work is of general interest to the community and certainly makes a nice contribution to the field which is certainly worthy of publication. However, claims have to be backed up by proper data, which in the eyes of this referee is still not the case here.

- In my earlier report, I raised the point that the key claim of this paper, namely the high E or Z selectivity in the transfer semihydrogenation is not backed up by proper data. The authors have responded to this by adding some crude spectra of the compounds I mentioned as examples (sic!) in my last report. So, for very few of the compounds now the ¹H NMR spectra are given. This is not the case for compounds 2c, 2e, 2f, 2h, 2k, 2l (here it should have been easy employing ¹⁹F NMR), 2m, 2q; resolution is not acceptable for 2d, 2j, 2o, 2p (here, the authors also miscalculated the ratios); for 2g a crude spectrum is given but it is completely unclear where the E/Z ratio comes from as no peaks are picked. This is where I stop.

I repeat my criticism that the main claim to high stereoselectivity is not backed up by proper data. Why is there no GC data? This would have been the standard way to report ratios of diastereomers. Furthermore, many of the high ratios as claimed by the authors have values which are outside of the instrumental preciseness of an NMR spectrometer. Again, GC data would have helped to solve this.

Response: Thanks again for the reviewer's comments. We have repeated the experiments for all substrates to check the selectivities by GC, which is indeed more precise to determine the

E/Z ratio. Generally, the results are in parallel with the NMR judgement. The corresponding GC spectra have been added to the SI, and the ratios in the manuscript have been revised accordingly. We will continue to use GC test in future studies in this field.

- Based on the spectral evidence as outlined above and extensively in my first report, it is not clear how the authors obtain pure spectra of the final alkenes. In their response to my first report, the authors have added a sentence to the SI that the diastereomers were separated by "simple filtration" (the authors do not give the filtering agent here). It is simply not possible to remove an unwanted E/Z-isomer by filtration unless special filtering agents such as Ag-coated silicagel has been employed. Furthermore, based on the experience in my group on catalytic alkyne (transfer) semihydrogenation, I claim based on my year-long experience that for most of the substrates presented here a separation via column chromatography is simply not possible. Therefore, it remains unclear how the authors come up with clean spectra such as the ones presented.

Response: Thanks for the comments. We are sorry for our misleading description, since simple filtration was employed to provide the crude mixture used for instrumental analysis to determine the selectivity but not to determine the selectivity. Exactly as the reviewer mentioned, the isomers are really difficult to separate due to their quite similar polarity. Complete separation is almost not possible in most cases. The figures below present the TLC and separation process. Generally, the polarity of the *Z*-alkenes is slightly less than the *E*-isomers. Extremely careful column chromatography was able to partially deliver the major product in a pure form. Take the transfer hydrogenation reaction of template substrate **1a** under the two standard conditions for example, the corresponding chromatography column is shown in Figure (d), with 300-400 mesh silica gel as the filler, petroleum ether as the eluent, and small test tubes were used to carry the eluent. The corresponding TLC plates are shown in Figure (f, g). We collected the two fractions of eluent separately (Figure (e, h)): to provide precise NMR spectra of the major products, the pure parts were collected and concentrated. Of course, the overall isolated yield was calculated based on the combination of all parts. The SI (pages 38 and 49) has been revised with corrected description and necessary details.

The authors seemingly disregard my earlier comment on the difference between transfer semihydrogenation and semihydrogenation and keep the title of their manuscript the same. In the eyes of this referee, there is certainly nice chemistry presented in this paper, as outlined above. However, when the key claims cannot be verified nor falsified, the paper does not hold to scientific standards. I therefore suggest the rejection of the manuscript.

Response: We are sorry for our misunderstanding of the reviewer's previous comment. The title has been modified to "Modulation of Metal Species as Early Control Point for Ni-catalyzed Stereodivergent *Transfer* Semihydrogenation of Alkynes with Water", and the corresponding descriptions in the manuscript have been checked through and corrected.

Thanks again for the reviewer's guiding comments, which remind us to be more rigorous in not only this work, but also in future research. We hope the revised manuscript would meet with the reviewer's criteria.

Reviewer #4

This work presents a Ni-catalyzed stereodivergent semihydrogenation of acetylenes with water. The strategy features use of cheap catalysts and nontoxic reagents, tolerance to a wide range of alkynyl substrates, and moderate to high yields of olefinic products. The authors proposed base modulates the valence state of active nickel species at an early stage, and consequently leads to Z- and E-isomers in disparate catalytic pathways: the in situ formed Ni(II) species delivers Z-alkenes, while the Ni(I) species affords E-alkenes as final products.

I think the study is of significance in both academic and practical perspectives, if the authors can offer more adequate computational data to justify their proposed mechanism.

Since no key intermediate could be isolated, as the authors stated, energy data based on calculation of key stationary points is a good alternate. However, from the view of theoretical study, it is not rigorous enough to identify the dominant reaction pathway only by comparison of the relative

energies of forked intermediates or products, some key transition states (TSs) should be located.

Response: Thanks for the reviewer's constructive comments, which led us to a more integral work. We are also grateful to the editor for the extension of the resubmission time. We have tried our best to locate the key transition states accordingly. Due to the complexity of nickel catalyst system, it is challenging to figure out the exact in-depth mechanism, since a variety of possibilities and factors (e.g. the participation of alkynes, alkenes, solvents, etc.) for each step would take effect. Guided by the professional suggestions, we are getting close to a more detailed mechanistic insight after substantial trials including cooperation with researchers on computational chemistry. Therefore, although not all the problems are unambiguously solved, at present, there is no formidable conflict between the theoretical studies and the mechanistic proposal, which is also supported by all the experimental results and existed reports. Although great efforts are still required and will be made in our laboratory, we hope the revision at this stage could meet with the publishing criteria.

*Based on the materials available, I think the authors want to address two important points about their catalytic cycles in Fig 2(e), that is, (1) no Ni(I) species (**J** or **J+Q** in Fig 2(f)) can be formed when PhCOONa is used, while Ni(I) species (**J** and **K**) is possible in the CF₃COONa case; (2) when PhCOONa is used, the intermediate **E** does not isomerize to **E'** (**E**-isomer of **E**), while intermediate **M** can isomerize to **N** in the CF₃COONa case. To rationalize these two points theoretically, the authors are suggested to conduct further calculations as follows:*

*(1) Locate TSs corresponding to **C****D**, **C****I**+**C**(or **J+Q**) and **B****C**, **B****P** processes in the left part of Fig 2(f), and exclude the possibility of **J** and **J+Q**'s formation when taking PhCOONa as base, through comparison of corresponding activation free energy barriers.*

Response: Thanks for the professional suggestion. Corresponding calculations have been conducted accordingly. Firstly, nickel benzoate **B** requires a 25.2 kcal/mol of activation free energy barrier to perform ligand exchange to **C**. Although we couldn't locate the transition state of further ligand exchange from **C** to **P**, the 29.9 kcal/mol of activation energy barrier from **P** to **Q** was unfavored. The energy barrier of **C****D** is 3.9 kcal/mol higher than **C****I**, while still easy to accomplish under the experimental temperature of 80°C. Unfortunately, the transition state for the subsequent comproportionation process to **J** or **J+Q** was not successfully located, which might be too high to proceed due to the intersystem crossing. We are very sorry for not being able to provide an explicit theoretical protocol at this stage. Considering that all experimental results including the EPR analysis and kinetic studies agreed with our proposal, we believe it is still the most reasonable pathway. The above results have been included in the revised manuscript and supplementary information.

(2) Locate TSs corresponding to **H**→**I** and **H+I**→**J+K** processes in the right part of Fig 2(e), and demonstrate the possibility of **J+K**'s formation when taking CF₃COONa as base, under an experimental temperature of 80 °C.

Response: Thanks for the suggestion. We tried to conduct the computational studies for the whole process. Firstly, the ligand exchange of nickel trifluoroacetate **G** to **H** requires a high activation free energy of 29.4 kcal/mol, which is in parallel with the experimental result that the reactivity was almost totally shut down at a lower temperature of 60 °C (Table 1, Entry 14). In contrast, the energy barrier for ligand exchange from **B** to **C** in the hydrogenation system with PhCO₂Na is lower (25.2 kcal/mol), so the reaction could still proceed smoothly at 60 °C (Table 2, Entry 11). For the subsequent process, the activation free energy barrier of reductive elimination from **H** to **I** is 14.5 kcal/mol, while the energetic requirement from the generation of Ni(II)-H species **R** is markedly higher. However, we failed in locating the transition state of the comproportionation step from **H+I** to **J+K**. We tried to search for literatures with similar computational studies for reference. However, despite the profound advance in relative area, we could hardly find instructive reports on the corresponding theoretical studies, which might be due to the complexity of the reaction systems with too many factors to be considered. We will keep focusing on the developed nickel catalyst system including further mechanistic investigations.

(3) Locate transition state (TS) corresponding to $H \rightarrow D'$ (replace PhCOO- group with CF₃COO- group in D) process in the right part of Fig 2(e), and exclude the possibility that intermediate **H** participates the catalytic cycle directly through transformation into intermediate **D'** when CF₃COONa is used, through comparison of corresponding activation free energy barriers.

Response: Thanks for the suggestion. The activation free energy barrier for the reaction of **H** with H₂O molecule en route to Ni(II)-H species **D'** (**R** in revised manuscript) is nearly 10 kcal/mol higher than that of the reductive elimination. Besides, **J+K** is lower than **R** in the potential energy surfaces, basically excluding the participation of **D'** in the catalytic cycle. The above DFT calculations have been added in Figure 3.

(4) Locate TS corresponding to $E \rightarrow E'$ process in the left part of Fig 2(f), and exclude the possibility of $E \rightarrow E'$ transformation when taking PhCOONa as base, through comparison of corresponding activation free energy barriers.

Response: Thanks for the constructive suggestion. We have located the free energy reaction profiles of the two reaction pathways. However, $E \rightarrow E'$ (S in supplementary information) transformation turned out to be also achievable, although the energy barrier is higher than the isomerization of $M \rightarrow N$, which might make it more predisposed to undergo hydrolysis directly once the vinyl nickel is formed. We also tried to calculate other possible processes including direct trans-addition of Ni-H species to alkenes as well as isomerization via nickel carbenoid intermediates (Chem. Sci., 2017, 8, 2914-2922; Org. Lett., 2021, 23, 5772-5776), but failed to locate the transition states. Taking consideration of the computational results, the supportive mechanistic experiments and also existing reports on isomerization of alkenyl Ni(I) species (Chem. Sci., 2016, 7, 5815-5820; Chem. Sci., 2020, 11, 10204-10211; Org. Lett. 2020, 22, 6982-6987, et al), the proposed catalytic cycles are still the most reasonable pathways. The results have been added in supplementary information Figure S5.

2. On page 15, in the SI file, the author stated, “The intrinsic reaction coordinate (IRC) calculations was also performed to verify the connectivity of the transition state and the energy minima”. However, I can’t find corresponding IRC results there or in the text. It’s suggested to be added at least into the SI file, in the right place.

Response: Thanks for the comments. The frequency analysis of the transition states gives only one imaginary frequency, which corresponds to the correct vibration mode. The correctness of all transition states has been verified with IRC calculations to connect the correct energy minima. The following figure shows the IRC graph of **TSCD** as an example, and the corresponding file is added to the supplementary information Figure S6.

Intrinsic Reaction Coordinate = 0, Total Energy (Hartree) = -1723.110454

3. On page 6, in the rebuttal file, the author stated, “However, activation of **C** with H₂O molecule en route to Ni(II)-H species **D** is more favored than the above process, supporting our mechanistic investigations and conjecture that the Z-selective catalytic cycle was initiated by [PhCOO-Ni(II)L-BPin] species”. I don't agree well with this statement. As shown in Fig 2(f) in the text, although **D** is relatively more stable when comparing with **J+Q**, this is not sufficient to rule out the possibility of **J+Q**'s formation from **C**, because both the formation of **D** and **J+Q** from **C** are obviously exothermic processes. Considering **J+Q** is ~40 kcal/mol lower than **C** in the potential energy surfaces, the reverse pathway (from **J+Q** to **C**) is nearly impossible under an experimental temperature of 80°C, that is, once **J+Q** is formed, it can no longer be transformed to **D**. So, in the absence of activation free energy barriers corresponding to transformations from **C** to **D** and to **I+C** (or **J+Q**), it is not rigorous enough to identify **D** as the only product (intermediate). In addition, **P** is slightly more stable than **C**, possible conversion of intermediate **B** to **J** via **P** should be excluded, too.

Response: Thanks very much for the professional comments. As mentioned in the reply to the first issue, although the proposed Ni(II) catalytic cycle has been proved to be energetically accommodated in the reaction conditions, the possibility from **C** to **J + Q** could not be definitely excluded just from the computational study, since we were not able to locate the transition state for comproportionation and ligand exchange from **C** to **P**. However, taking consideration of all the experimental results including the EPR analysis, the control experiments and the kinetic studies, our mechanistic proposal is still highly supported and reasonable. Further mechanistic explorations including capturing of the intermediates and systematic computational studies will be conducted in our laboratory.

4. If the authors submit a new version, it is recommended to add different symbols onto the potential energy profiles in Fig 2(f), so that readers can distinguish different reaction pathways easily in the printed version.

Response: Thanks for the kind suggestion, new symbols have been added in the revised manuscript and supplementary information to make it more readable.

Thanks again for all of the professional comments, which truly help us to make a more comprehensive work. We have tried our best to answer all the questions, but there are indeed some computational issues difficult to solve explicitly. The manuscript has been revised according to the results, with some parts presented in SI, just for the readers' reference and avoiding confusion in the paper. Despite that sophisticated work is still awaiting to uncover the deep insights of the reaction system, the proposed mechanism is in accordance with current experimental and theoretical studies. We believe the novelty and academic value in our research will impress the community and make a nice contribution to the field. We hope the revised manuscript is suitable for publication in *Nature Communications*. We will keep focusing on further pursuit including the development of the catalytic strategy and also understanding of the detailed mechanisms.

REVIEWER COMMENTS

Reviewer #4 (Remarks to the Author):

This paper is a revised manuscript previously submitted to Nature Communications as a full paper. The authors have added substantial computational trials and located most of the key transition states according to my initial comments, although encountered failure in a few transition states' location. Now the proposed mechanism looks more credible, benefited from synergic supports of experimental and theoretical results. I think the present manuscript is suitable for the publication as it stands.

Reviewer #5 (Remarks to the Author):

Stereodivergent alkyne semihydrogenation was the subject of studies by several scientific groups. The stereoselectivity control was achieved by alteration of the catalyst, ligand, solvent, or metal-to-ligand ratio. In most cases, (E)-alkenes were obtained in (Z)→(E) isomerization of the previously produced (Z)-alkenes.

The authors presented a methodology that allows the stereoselective synthesis of (Z)- or (E)-alkene depending on the base used. Interestingly, (Z)→(E) isomerization of (Z)-alkene was not observed under the reaction conditions. The applicability of both variants was proved by a wide spectrum of substrates tested.

The authors responded adequately to the concerns in the previous round of review. Diastereomeric ratios were calculated based on GC analysis. Necessary spectroscopic data was provided. However, the sentence "We are sorry for our misleading description, since simple filtration was employed to provide the crude mixture used for instrumental analysis to determine the selectivity but not to determine the selectivity." is unclear.

The most important part of the work is mechanistic studies, some aspects of which should be clarified.

The article may be indeed published after addressing the following questions:

1) (Figure 1a) If D₂O is a sole hydrogen donor, why D/H ratio is low (e.g., 60/40)?

Zhou and Fan (10.1039/C9CC01970G) previously reported on deuterium incorporation in the benzene ring.

Did the authors observe the same phenomenon? If so, the observation should be reflected in the mechanism.

2) (Figure 1a) What is the KIE in the (Z)- or (E)-selective variant? Please comment on the obtained results in the manuscript.

3) (Figure 1b) Most importantly, Huang and Luo (10.1021/acsomega.1c01083) observed the dependence of B2pin2 amount on vinyl boronate protodeborylation in a similar catalytic system. The application of the standard conditions (1 equiv. of B2pin2) gave no product in contrast to lower quantities of B2pin2.

Similarly, Shi et al. (10.1039/C9CC03213D) used a smaller amount of B2nep2) to obtain (Z)-alkene from vinyl boronate.

Based on these findings, the authors of both papers included vinyl boronate species as intermediates in the catalytic cycle. Unsuccessful reactions 9-12 may be caused by B2pin2 excess (3 equiv.). Please study this relationship (e.g., 0, 1, 2 equiv. of B2pin2) regarding protodeborylation reactions 9-12 and revise the mechanism if necessary.

4) The temperature affected (Z)→(E) isomerization (Supplementary Scheme 1) under A conditions. How does the decrease in temperature (e.g., 40 °C) influence the stereoselectivity of alkyne 1a semihydrogenation under B conditions?

5) The authors postulated the production of hydride species in both catalytic cycles. Is there any spectroscopic evidence of their formation?

6) Some references are missing.

Papers on stereodivergent semihydrogenation of the C–C triple bond:

Shen, R.; Chen, T.; Zhao, Y.; Qiu, R.; Zhou, Y.; Yin, S.; Wang, X.; Goto, M.; Han, L.-B. Facile Regio- and Stereoselective Hydrometalation of Alkynes with a Combination of Carboxylic Acids and Group 10 Transition Metal Complexes: Selective Hydrogenation of Alkynes with Formic Acid. *J. Am. Chem. Soc.* 2011, 133 (42), 17037–17044. <https://doi.org/10.1021/ja2069246>.

Kusy, R.; Grela, K. E- and Z-Selective Transfer Semihydrogenation of Alkynes Catalyzed by Standard Ruthenium Olefin Metathesis Catalysts. *Org. Lett.* 2016, 18 (23), 6196–6199. <https://doi.org/10.1021/acs.orglett.6b03254>.

Yang, J.; Wang, C.; Sun, Y.; Man, X.; Li, J.; Sun, F. Ligand-Controlled Iridium-Catalyzed Semihydrogenation of Alkynes with Ethanol: Highly Stereoselective Synthesis of E- and Z-Alkenes. *Chem. Commun.* 2019, 55 (13), 1903–1906. <https://doi.org/10.1039/C8CC09714C>.

Huang, Z.; Wang, Y.; Leng, X.; Huang, Z. An Amine-Assisted Ionic Monohydride Mechanism Enables Selective Alkyne Cis-Semihydrogenation with Ethanol: From Elementary Steps to Catalysis. *J. Am. Chem. Soc.* 2021, 143 (12), 4824–4836. <https://doi.org/10.1021/jacs.1c01472>.

The recent examples may be added as well:

Kusy, R.; Lindner, M.; Wagner, J.; Grela, K. Ligand-to-Metal Ratio Controls Stereoselectivity: Highly Functional Group-Tolerant, Iridium-Based, (E)-Selective Alkyne Transfer Semihydrogenation. *Chem Catal.* 2022, 0 (0). <https://doi.org/10.1016/j.checat.2022.04.014>.

Luo, J.; Liang, Y.; Montag, M.; Diskin-Posner, Y.; Avram, L.;

Milstein, D. Controlled Selectivity through Reversible Inhibition of the Catalyst: Stereodivergent Semihydrogenation of Alkynes. *J. Am. Chem. Soc.* 2022, 144 (29), 13266–13275. <https://doi.org/10.1021/jacs.2c04233>.

Papers on semihydrogenation of the C–C triple bond using B2pin2:

He, G.; Zhang, Q.; Huang, H.; Chen, S.; Wang, Q.; Zhang, D.; Zhang, R.; Zhu, H. Copper(I)-Catalyzed Highly Regio- and Stereoselective Boron Addition–Protonolysis of Alkynamides to Give Alkenamides. *Eur. J. Org. Chem.* 2013, 2013 (30), 6979–6989. <https://doi.org/10.1002/ejoc.201300947>

Huang, J.; Li, X.; Wen, H.; Ouyang, L.; Luo, N.; Liao, J.; Luo, R. Substrate-Controlled Cu(OAc)₂-Catalyzed Stereoselective Semi-Reduction of Alkynes with MeOH as the Hydrogen Source. *ACS Omega* 2021, 6 (17), 11740–11749. <https://doi.org/10.1021/acsomega.1c01083>.

Dear Sir/Madam,

On behalf of my co-authors, thanks again for the positive and constructive comments and suggestions on our manuscript entitled “Modulation of Metal Species as Early Control Point for Ni-catalyzed Stereodivergent Transfer Semihydrogenation of Alkynes with Water”. We have studied the comments carefully and made corresponding corrections which we hope would meet with approval. Revised parts are yellow marked in the paper. A point by point response to the comments are listed as follows.

Reviewer #4 (Remarks to the Author):

This paper is a revised manuscript previously submitted to Nature Communications as a full paper. The authors have added substantial computational trials and located most of the key transition states according to my initial comments, although encountered failure in a few transition states' location. Now the proposed mechanism looks more credible, benefited from synergic supports of experimental and theoretical results. I think the present manuscript is suitable for the publication as it stands.

Response: Thanks for the reviewers for their affirmation. The guiding comments from them helped us a lot to improve the manuscript.

Reviewer #5 (Remarks to the Author):

Stereodivergent alkyne semihydrogenation was the subject of studies by several scientific groups. The stereoselectivity control was achieved by alteration of the catalyst, ligand, solvent, or metal-to-ligand ratio. In most cases, (E)-alkenes were obtained in (Z)→(E) isomerization of the previously produced (Z)-alkenes.

The authors presented a methodology that allows the stereoselective synthesis of (Z)- or (E)-alkene depending on the base used. Interestingly, (Z)→(E) isomerization of (Z)-alkene was not observed under the reaction conditions. The applicability of both variants was proved by a wide spectrum of substrates tested.

The authors responded adequately to the concerns in the previous round of review. Diastereomeric ratios were calculated based on GC analysis. Necessary spectroscopic data was provided. However, the sentence “We are sorry for our misleading description, since simple filtration was employed to provide the crude mixture used for instrumental analysis to determine the selectivity but not to determine the selectivity.” is unclear.

Response: We are awfully sorry for our carelessness. We were trying to state that simple filtration was employed to remove the insoluble impurities to make the crude mixture ready for instrumental analysis to determine the selectivity, but not for the separation of the Z- and E-isomers.

The most important part of the work is mechanistic studies, some aspects of which should be clarified.

The article may be indeed published after addressing the following questions:

1) (Figure 1a) If D₂O is a sole hydrogen donor, why D/H ratio is low (e.g., 60/40)? Zhou and Fan (10.1039/C9CC01970G) previously reported on deuterium incorporation in the benzene ring.

Did the authors observe the same phenomenon? If so, the observation should be reflected in the mechanism.

Response: Thanks very much for the constructive suggestion. As for the low deuterium isotopic content in the indicated reaction, tiny amounts of water in the system might be taking effect. We tried to repeat the reaction more rigorously with dried reagents, and the D/H ratio of **2a'** reached 71/29. To explicitly check the deuterium incorporation in the benzene ring, we performed the reduction of diphenylacetylene **1g** with D₂O to compare with Zhou and Fan's work. As shown below, 68% of deuteration at the vinylic positions was observed, without deuteration in the benzene ring. The corresponding results have been added in the revised manuscript and supplementary information.

2) (Figure 1a) What is the KIE in the (Z)- or (E)-selective variant? Please comment on the obtained results in the manuscript.

Response: Thanks for the suggestion. We have supplemented the corresponding kinetic

experiments. The kinetic isotopic effect ($k_H/k_D = 1.48$) was observed when H₂O was replaced by D₂O in the *Z*-selective reactant stream (Figure a). A kinetic isotopic effect of 1.06 was obtained in the *E*-selective reduce system (Figure b). These results indicated that activation of H₂O molecule delivering Ni-H species might not be involved in the rate-determining step in both conditions. The KIE studies have been added in the revised manuscript and supplementary information.

3) (Figure 1b) Most importantly, Huang and Luo (10.1021/acsomega.1c01083) observed the dependence of B_2pin_2 amount on vinyl boronate protodeborylation in a similar catalytic system. The application of the standard conditions (1 equiv. of B_2pin_2) gave no product in contrast to lower quantities of B_2pin_2 :

Scheme 2. Effect of Various Equivalents of B_2pin_2 for the Transformation of Possible Intermediates

Similarly, Shi et al. (10.1039/C9CC03213D) used a smaller amount of B_2nep_2 to obtain (*Z*)-alkene from vinyl boronate:

Based on these findings, the authors of both papers included vinyl boronate species as intermediates in the catalytic cycle. Unsuccessful reactions 9-12 may be caused by B_2pin_2 excess (3 equiv.). Please study this relationship (e.g., 0, 1, 2 equiv. of B_2pin_2) regarding protodeborylation reactions 9-12 and revise the mechanism if necessary.

Response: Thanks very much for the professional comments. The corresponding control experiments have been supplemented. Vinyl boronates **4**, **5** and diborylated vinyl derivatives **4'**, **5'** were respectively performed under the both reaction conditions with 0, 1.0, 2.0 equiv. of B_2pin_2 . Olefin products **2g** and **3g** were not detected (equations (9), (10), (11) and (12)), further excluding the possibility for the protodeborylation of vinyl boronate species. These results have been added in the revised manuscript (Figure 1b).

4) The temperature affected (*Z*)→(*E*) isomerization (Supplementary Scheme 1) under A conditions. How does the decrease in temperature (e.g., 40 °C) influence the stereoselectivity of alkyne **1a** semihydrogenation under B conditions?

Response: Thanks for the comments. The corresponding control experiment under condition B at lower temperature of 40 °C has been supplemented. The alkyne was untouched in this condition, leaving the starting materials recovered. The reaction at 60 °C afforded the *Z*-alkene product in 4% yield without the formation of the *E*-alkene. The corresponding results have been added in the revised manuscript (Table 1, entries 14 and 15).

^a**1a** (0.15 mmol), NiBr₂ (5 mol%), 2,2'-bipyridine (11 mol%), CF₃CO₂Na (2.0 equiv.), B₂Pin₂ (3.0 equiv.), H₂O (3.0 equiv.), DMF (2 mL), 40 °C or 60 °C in Ar atmosphere for 24 h.

5) The authors postulated the production of hydride species in both catalytic cycles. Is there any spectroscopic evidence of their formation?

Response: Thanks very much for the comments. Previously, we tried *in situ* HRMS, but no supportive information was obtained. As a continued effort, *in situ* NMR was conducted to monitor nickel species. Unfortunately, as shown below, no peak for Ni-H or other nickel species was observed for all the reactions with both catalytic and equivalent amount of catalysts. So far, the proposed mechanism with nickel hydride species is reasonable based other mechanistic insights as well as existing reports on transition metal catalyzed transfer hydrogenation with water as hydrogen source using diboron compounds (J. Am. Chem. Soc. 138, 6107-6110 (2016); Org. Lett. 18, 4250-4253 (2016); Angew. Chem. Int. Ed. 56, 3987-3991 (2017), et al). We will keep focusing on the hydrogen transfer system including mechanistic studies in the future.

6) Some references are missing.

Papers on stereodivergent semihydrogenation of the C–C triple bond: Shen, R.; Chen, T.; Zhao, Y.; Qiu, R.; Zhou, Y.; Yin, S.; Wang, X.; Goto, M.; Han, L.-B. Facile Regio- and Stereoselective Hydrometalation of Alkynes with a Combination of Carboxylic Acids and Group 10 Transition Metal Complexes: Selective Hydrogenation of Alkynes with Formic Acid. *J. Am. Chem. Soc.* 2011, 133 (42), 17037–17044. <https://doi.org/10.1021/ja2069246>.

Kusy, R.; Grela, K. E- and Z-Selective Transfer Semihydrogenation of Alkynes Catalyzed by Standard Ruthenium Olefin Metathesis Catalysts. *Org. Lett.* 2016, 18 (23), 6196–6199. <https://doi.org/10.1021/acs.orglett.6b03254>.

Yang, J.; Wang, C.; Sun, Y.; Man, X.; Li, J.; Sun, F. Ligand-Controlled Iridium-Catalyzed Semihydrogenation of Alkynes with Ethanol: Highly Stereoselective Synthesis of E- and Z-Alkenes. *Chem. Commun.* 2019, 55 (13), 1903–1906. <https://doi.org/10.1039/C8CC09714C>.

Huang, Z.; Wang, Y.; Leng, X.; Huang, Z. An Amine-Assisted Ionic Monohydride Mechanism Enables Selective Alkyne Cis-Semihydrogenation with Ethanol: From Elementary Steps to Catalysis. *J. Am. Chem. Soc.* 2021, 143 (12), 4824–4836. <https://doi.org/10.1021/jacs.1c01472>.

The recent examples may be added as well: Kusy, R.; Lindner, M.; Wagner, J.; Grela, K. Ligand-to-Metal Ratio Controls Stereoselectivity: Highly Functional Group-Tolerant, Iridium-Based, (E)-Selective Alkyne Transfer Semihydrogenation. *Chem Catal.* 2022, 0 (0). <https://doi.org/10.1016/j.checat.2022.04.014>.

Luo, J.; Liang, Y.; Montag, M.; Diskin-Posner, Y.; Avram, L.; Milstein, D. Controlled Selectivity through Reversible Inhibition of the Catalyst: Stereodivergent Semihydrogenation of Alkynes. *J. Am. Chem. Soc.* 2022, 144 (29), 13266–13275. <https://doi.org/10.1021/jacs.2c04233>.

Papers on semihydrogenation of the C–C triple bond using B₂pin₂: He, G.; Zhang, Q.; Huang, H.; Chen, S.; Wang, Q.; Zhang, D.; Zhang, R.; Zhu, H. Copper(I)-Catalyzed Highly Regio- and Stereoselective Boron Addition–Protonolysis of Alkynamides to Give Alkenamides. Eur. J. Org. Chem. 2013, 2013 (30), 6979–6989. <https://doi/abs/10.1002/ejoc.201300947>

Huang, J.; Li, X.; Wen, H.; Ouyang, L.; Luo, N.; Liao, J.; Luo, R. Substrate-Controlled Cu(OAc)₂-Catalyzed Stereoselective Semi-Reduction of Alkynes with MeOH as the Hydrogen Source. ACS Omega 2021, 6 (17), 11740–11749. <https://doi.org/10.1021/acsomega.1c01083>.

Response: Thanks very much for the suggestions. The mentioned references have been cited in the revised manuscript (references 11–14, 21, 22, 45, 48).

Thanks again for all of the professional comments, which truly lead us to make a more comprehensive work. We hope the revised manuscript is suitable for publication in *Nature Communications*.

REVIEWERS' COMMENTS

Reviewer #5 (Remarks to the Author):

The authors responded adequately to the concerns of the previous round of review.

However, some reservations were raised regarding the KIE experiments. The authors stated that the low D/H ratio while using D₂O was due to “tiny amounts of water”, the presence of which in the system obviously affects the KIE results. Thus, I would suggest to focus again on it and perhaps consider reviewing/altering/commenting these KIE experiments in question, because their outcomes may be misleading (at least for me they are).

Nevertheless, the rest is perfect, the chemistry is sound, so the manuscript shall be considered for publication.

Dear Sir/Madam,

On behalf of my co-authors, thanks again for the positive and constructive comments and suggestions on our manuscript entitled “Modulation of Metal Species as Control Point for Ni-catalyzed Stereodivergent Semihydrogenation of Alkynes with Water”. We have studied the comments carefully and made corresponding corrections which we hope would meet with approval. Revised parts are yellow marked in the paper. The responses to the comments are listed as follows.

Reviewer #5 (Remarks to the Author):

The authors responded adequately to the concerns of the previous round of review.

However, some reservations were raised regarding the KIE experiments. The authors stated that the low D/H ratio while using D₂O was due to “tiny amounts of water”, the presence of which in the system obviously affects the KIE results. Thus, I would suggest to focus again on it and perhaps consider reviewing/altering/commenting these KIE experiments in question, because their outcomes may be misleading (at least for me they are).

Nevertheless, the rest is perfect, the chemistry is sound, so the manuscript shall be considered for publication.

Response: Thanks very much for the constructive suggestion. In the last revision, we tried to repeat the deuterium labeling experiments as rigorously as we could, failing to further improve the deuterium isotopic content of alkene products. In view of the effects of H₂O on the deuterium labeling experiments, the KIE results were calibrated by D/H ratios. Correction are as follows:

Figure 1. The Kinetic Isotopic Effect of Z-selective Hydrogenation of **1a**

Unadjusted result of kinetic isotopic effect:

$$\text{KIE}' = k_H'/k_D' = 0.32333/0.219 = 1.48$$

Calibrated result of kinetic isotopic effect:

The D/H ratios of deuterium-labeled products all reached 80/20 in Z-selective reactions.

$$(100\% k_H)/(80\% k_D+20\% k_H) = 0.32333/0.219$$

$$\text{KIE} = k_H/k_D = 1.68$$

Figure 2. The Kinetic Isotopic Effect of *E*-selective Hydrogenation of **1a**

Unadjusted result of kinetic isotopic effect:

$$\text{KIE}' = k_H'/k_D' = 0.2581/0.24433 = 1.06$$

Calibrated result of kinetic isotopic effect:

The D/H ratios of deuterium-labeled alkenes were observed to be 70/30 in *E*-selective reactions.

$$(100\% k_H)/(70\% k_D + 30\% k_H) = 0.2581/0.24433$$

$$\text{KIE} = k_H/k_D = 1.08$$

Calibrated result of kinetic isotopic effect in *Z*-selective reactant stream was calculated to be 1.68. The actual KIE value is estimated to be between 1.48 and 1.68 due to the ambiguous effects of water on deuterium labeling experiments. And it is obvious that no significant KIE is observed in either conditions. The corresponding discussion has been added in the revised manuscript and supplementary information.

Thanks again for the professional comments.